# A new approach to measuring absolute pitch on a psychometric theory of isolated pitch perception: Is it disentangling specific groups or capturing a continuous ability?

**Nayana Di Giuseppe Germano**[1]*, **Hugo Cogo-Moreira**[2], **Fausto Coutinho-Lourenço**[3], **Graziela Bortz**[4]

**1** Department of Music, Federal University of Santa Maria, Santa Maria, RS, Brazil, **2** School of Public Health, The University of Hong Kong, Hong Kong, SAR, China, **3** Department of Psychobiology, Federal University of Sao Paulo, Sao Paulo, SP, Brazil, **4** Department of Music, UNESP, Sao Paulo, SP, Brazil

☯ These authors contributed equally to this work.
* nayana.germano@ufsm.br

**Data Availability Statement:** All relevant data are within the paper and its Supporting Information files.

## Abstract

Absolute Pitch (AP) is commonly defined as a rare ability that allows an individual to identify any pitch by name. Most researchers use classificatory tests for AP which tracks the number of isolated correct answers. However, each researcher chooses their own procedure for what should be considered correct or incorrect in measuring this ability. Consequently, it is impossible to evaluate comparatively how the stimuli and criteria classify individuals in the same way. We thus adopted a psychometric perspective, approaching AP as a latent trait. Via the Latent Variable Model, we evaluated the consistency and validity for a measure to test for AP ability. A total of 783 undergraduate music students participated in the test. The test battery comprised 10 isolated pitches. All collected data were analyzed with two different rating criteria (*perfect* and *imperfect*) under three Latent Variable Model approaches: continuous (Item Response Theory with two and three parameters), categorical (Latent Class Analysis), and the Hybrid model. According to model fit information indices, the *perfect* approach (only exact pitch responses as correct) measurement model had a better fit under the trait (continuous) specification. This contradicts the usual assumption of a division between AP and non-AP possessors. Alternatively, the categorical solution for the two classes demonstrated the best solution for the *imperfect* approach (exact pitch responses and semitone deviations considered as correct).

## Introduction

The phenomenon of Absolute Pitch (AP) was first scientifically described by Stumpf [1], although it was alluded to much earlier in Mozart's era [2, 3]. AP ability has attracted attention from musicians, psychologists, and neuroscientists, leading to a large body of research [4–7].

AP has not yet been accurately and consensually defined among the academic community [8], leading to significant variations among AP evidence and AP classification. Consequently,

**Funding:** Germano, N: FAPESP 2016/08377-4 (Fundação de Amparo à Pesquisa do Estado de São Paulo) provided funding for the research and publication of this article. Cogo-Moreira, H: CAPES (Thesis award) Grant no. 0374/2016, process no. 23038.009191/2013-76) and CAPES-Alexander von Humboldt senior research fellowship (Grant 88881.145593/2017-01). Bortz, G: FAPESP 2019/02133-4 (Fundação de Amparo à Pesquisa do Estado de São Paulo) provided funding for the publication of this article.

**Competing interests:** The authors have declared that no competing interests exist.

conclusions regarding AP classification may not be comparable due to the lack of consensus on criteria (e.g., the time required to identify a tone, or the degree of precision in tone identification).

Only a few defining criteria for AP ability are agreed upon among authors, such as the automatic association between a certain pitch and a learned verbal label [9], and the definition of AP as a rare ability that refers to a long-term internal representation for pitches. Consequently, AP typically manifests behaviorally as the ability to identify any given pitch by name (according to the traditional pattern of musical notation learned by a subject), or by producing a given musical tone on demand, with no external reference, e.g., without a diapason [3, 10–13].

The extant AP literature references certain limitations of AP possessors in pitch identification. Timbre limitation is mentioned by several studies [1, 6, 14–18], although it has not yet been universally specified or quantified, and its causes have not yet been scientifically explained. The same can be said of register limitation among AP individuals [1, 14, 19, 20]. We consider these to be examples of relevant non-consensual criteria excluded from most AP definitions. An important consequence of this methodological decision is that individuals with difficulties in tone recognition due to certain configurations of musical parameters (mostly regarding timbre and/or register) must be considered as non-AP possessors.

The AP phenomenon is generally considered as instantaneous pitch recognition and some studies adopt a brief time response window (e.g., three or four seconds), assuming it is sufficient to affirm an immediate response in a given task [21, 22]. It is also assumed that providing some procedures in AP tests can limit (or completely eliminate) the use of Relative Pitch (RP). These procedures can include some methodological issues, including granting brief time response, separating tones by an interval larger than an octave, or placing brown noise between stimuli [16, 23, 24].

There are two different theoretical perspectives regarding the RP definition, namely, the broad and the narrow perspectives [4, 5]. In the broad perspective, RP ability is assigned to anyone (musician or non-musician) who is capable of realizing basic music perceptual tasks, such as recognizing familiar music when it is transposed or played on different instruments or singing a familiar song in tune [25]. These are predominantly intuitive unconscious abilities, and most people accomplish them instinctively. In the narrow perspective, RP is assigned to individuals who can name intervals and other musical elements (including triads, tonalities, harmonic progressions, and scales, among others). Musicians must be able to recognize familiar music, like non-musicians, and also aurally recognize and name basic musical elements used in compositions (e.g., whether the heard musical interval was a minor or major second) [26]. Hence, RP in the narrow perspective is acquired through years of intense training.

Thus, the use of classificatory tests to separate AP possessors from RP possessors should be approached with caution. Given that the study of music perception in undergraduate music schools and conservatoires includes sight-singing and ear training, these goals encompass the development of the RP ability among all students. Since most participants in AP studies are musicians, they all have received some degree of training in music perception. Thus, it is reasonable to expect that all test participants possess some degree of RP ability, even AP possessors. Consequently, it is impossible to ascertain whether RP ability can be completely eliminated by the use of a short response time or any other methodology. In fact, the possibility that an individual may possess both abilities, i.e., that AP and RP phenomena are not mutually exclusive, must be considered [5, 27, 28].

As posited by Levitin and Rogers [29], "AP is neither 'absolute' nor 'perfect' in the ordinary uses of those words". AP possessors not only exhibit limitations for timbre and registers, as mentioned in previous paragraphs, but they also frequently make octave and semitone errors. This occurs so commonly that a substantial portion of classificatory tests for AP consider

semitone errors as correct (or partially correct) answers [16, 21, 22, 30–34]. This leads to a core methodological issue found in AP literature, that is, a lack of agreement for criteria regarding cut-offs in AP classificatory tests, which are arbitrarily defined. Moreover, this affects what is considered a correct or an incorrect answer to the stimuli. An example can be observed in Dohn et al. [30], who used a pitch identification test described and provided by Athos et al. [22], which was originally developed by Baharloo et al. [21]. Although all three studies adopted the same test, they did not apply exactly the same methodology, nor the same scoring criteria. This lack of common criteria or a gold-standard tool to measure the same phenomenon leads to difficulties in comparing results, even when researchers intend to utilize the same test.

We aimed to develop a test for isolated pitch recognition from a psychometric perspective. That is, we considered this ability to be a latent phenomenon, evaluating a) the best model solution underlying the isolated pitch recognition, and b) how different rating approaches commonly used in the literature might influence the decision of the best model for isolated pitch recognition tasks. The use of a latent approach elucidates the item level functioning, providing evidence for construct validity.

## Materials and methods

### Participants and study design

A total of 783 undergraduate music students (n = 512; male = 65.4%) were recruited to this study. Participants ranged from the first to tenth semesters of study at seven different Brazilian universities, five of which were located in São Paulo city, and two in Curitiba city. This study was approved by the relevant ethics board (Ethics Committee's Approval CAAE: 60855816.3.0000.5477) and participants written consent was provided by all the students before the test/evaluation. The study was conducted during the first semester of the 2017 academic year. The participants' mean age was 24.7 years (range = 17 to 72) and they had an average of 10.29 years of music practice (SD = 6.7; range = 1 to 65). All participants were exposed to ear training and sight singing classes during their music studies.

The perception task consisted of five different batteries: isolated pitches, melodic intervals, harmonic intervals, fundamental position triads, and first position triads. Each battery included 10 stimuli. In this study, only the first battery (isolated pitches) will be discussed, which is the common procedure used to track AP. It must be emphasized that we considered the isolated pitch recognition without reference as a latent trait, without the automatic assumption that this ability and the AP ability were the same. Therefore, the items intended to measure the ability to identify isolated pitches without a reference was our main priority.

The first author of this study collected all data, giving exactly the same instructions to all subjects and guaranteeing an adequate standardization of method. The protocol was applied collectively, with previous authorization obtained from professors and the legal guardian responsible for each institution. Each stimulus was played once for 3 seconds, with a 15-second pause in between. No reference pitch was provided. Pitches and registers were highly variable among items. Timbres were chosen to represent each family of musical instruments. The stimuli were recorded in a studio by a professional and played on CD during the tests.

We attempted to limit the use of RP in our test by not providing any reference pitch, playing each stimulus only once, and changing the timbre and register between each stimulus. However, due to the issues discussed in the introduction section, we considered that it is methodologically impossible to completely prevent the use of RP in any isolated pitch recognition task. Each participant has a unique way of identifying pitches which can employ a combination of AP and RP, and common isolated pitch recognition tasks are incapable of evaluating the underlying mechanism being used. Consequently, we did not evaluate reaction time,

providing a 15-second window between each stimulus in our test. With a longer response time, subjects had sufficient time to look at the response sheet, choose their answer, confirm where the right response was located on the drawn piano-keyboard, and mark their answer. This decreased the chance of errors unrelated to pitch discrimination. Notably, all participants in this research were required to pass an aural skill test to be admitted to music programs in Brazilian Universities. This indicates that all participants received some degree of training in music perception. Thus, it is reasonable to expect that all participants possessed some degree of RP ability, even AP possessors.

Participants were instructed to indicate the pitch they thought was correct on a piano-key-board drawn on paper. It was expected that some participants would not be fluent in reading and/or writing traditional musical notations if they had not yet taken the appropriate courses. The drawn piano-keyboard allowed us to delimit specifically 12 possible answers (12 chromatic notes). The piano-keyboard was also chosen because it allowed for easier visualization and identification of each possible answer through key position. It also contained the verbally written note names.

Participants were informed that the first battery was composed of only isolated pitches. They were also informed about the duration of each stimulus and the time interval between them. This was necessary to avoid confusion and surprise among subjects. No information was provided regarding timbre. The drawn keyboard had only one octave, as the object of our study was pitch class recognition. Therefore, the octave parameter was not considered in this task, and was disregarded by all subjects.

The first battery contained 10 isolated pitches in 5 different timbres (piano, violin, flute, tuba, and voice). The voice was recorded from two professional vocalists in a studio. All other instruments were recorded with the *software Kontakt*, using professionally recorded samples. The piano was taken from *Piano in 162*, the violin from *Spitfire Solo Strings*, the flute from *8dio Claire Flute*, and the tuba from *Spitfire Symphonic Brass* (Fig 1).

In (Fig 1), the circle represents the latent trait, which we referred to as the ability to identify isolated pitches without reference (AIPWR). Because we were unable to measure any latent trait directly, the 10 stimuli constituted a set of items that could be measured and tested directly. These items, represented by rectangles, are similar to symptoms of psychological disorders, which can be directly observed. The stimuli are composed of three tone dimensions: register, timbre, and pitch class. We chose these 10 items to correspond to a summary of a vast stimuli range that is commonly used to measure AP ability. Thus, they were purposely very heterogeneous stimuli, encompassing all the different ranges of timbre, pitch, and register necessary to access Isolated Pitch Recognition Ability.

## Data analysis

To evaluate the psychometric features of the isolated pitches battery, we used Mplus version 8.0 [35] and the R program [36]. All collected data were analyzed under three approaches: continuous, categorical, and hybrid (the factor mixture model). The former Item Response Theory (IRT) approach assumes that there is a continuous latent measure (or "trait") underlying the 10 items. That is, each participant would have some ability to identify isolated pitches without reference, similarly to other continuous cognitive measurements like quotient intelligence, psychopathology, and language skills. Two different IRT models were used:

a. An IRT model with two parameters for each stimulus: the discrimination parameter (*parameter a*), which describes the ability of the stimuli to distinguish between persons with low and high pitch identification ability; and the item location parameter (*parameter b*), representing the level of pitch identification ability where there is a 50% chance of correctly identifying the pitch of the stimulus;

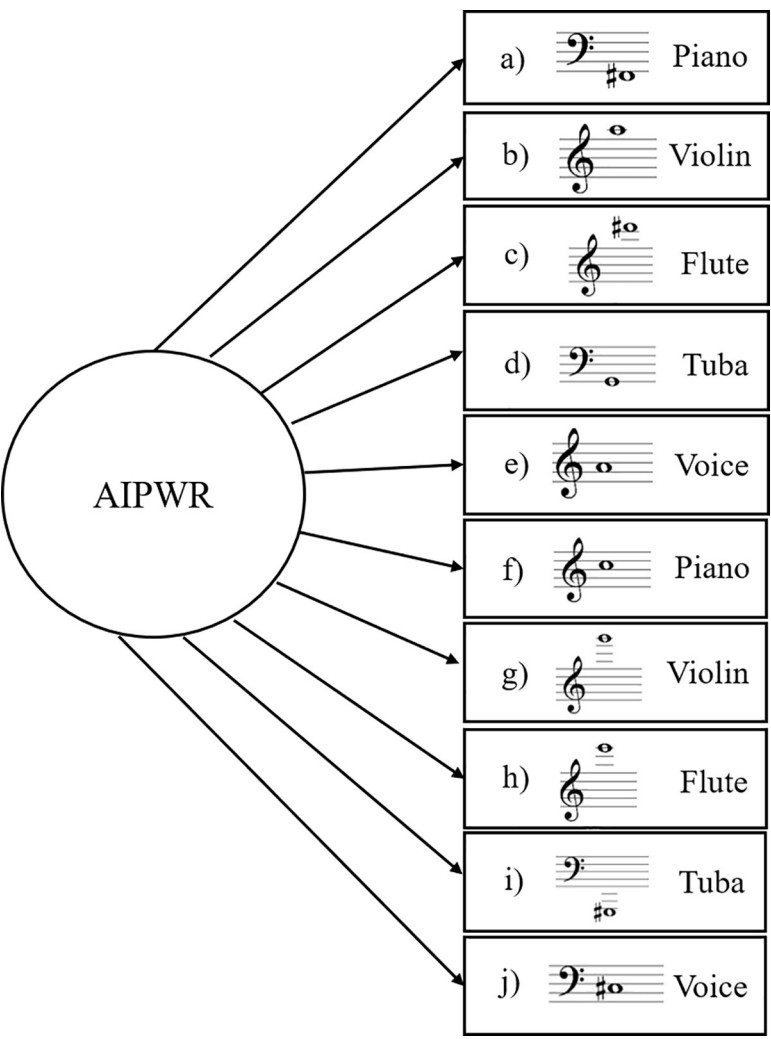

**Fig 1. Theoretical Model for isolated pitch recognition trait.** Theoretical Model proposed for Latent Trait AIPWR: Ability to identify Isolated Pitch Without Reference. Items are 10 isolated pitches in various timbres and registers without reference (*a-j*). The arrows indicate the ability from latent trait to items. Figure adapted from Germano et al. [8].

b. An IRT with three parameters (discrimination, difficulty, and guessing [aka. lower bound asymptote]), where the additional parameter, guessing, is the probability of a person with very low pitch identification ability still correctly providing a correct answer for a given stimulus. The guessing parameter was recently implemented in Mplus and uses a prior maximum likelihood parameter that helps in the convergence of the model [37].

We used a piano keyboard to track the answers of the participants. Out of 12 keys, participants chose only one. Therefore, the prior guessing parameter likelihood was 1/12 for the *perfect* rating and 3/12 for the *imperfect* rating criteria. According to Baker [38], discrimination parameter cutoffs are 0 (none); 0.1 to 0.34 (very low); 0.35 to 0.64 (low); 0.65 to 1.34 (moderate); 1.35 to 1.69 (high); < 1.70 (very high) and + infinity (perfect).

For IRT analysis, we used Maximum Likelihood estimator and logit parameterization (theta). The factor is assumed to be normally distributed being the mean fixed at zero and factor variance at 1. That is, the IRT analysis is centered on the person sample being at 0 logits,

and the item difficulty parameters are provided relative to this. For the difficulty parameter, values closer to 3 indicated more difficulty, and values closer to -3 indicated less difficulty. Values around zero indicated the middle point between both extremes. To evaluate the model fit indices for IRT models, a Pearson chi-square test for categorical outcomes was used, with *p*-values higher than 0.05 being indicators of a good fit. Item level fit was evaluated via Pearson's X2 (S-X2) implemented in R package mirt, as per Orlando and Thissen [39].

For the categorical approach, we used Latent Class Analysis (LCA), which classifies subpopulations where population membership is inferred from the data. LCA has some similarities with the *prima prattica* of AP research, where subjects are classified in homogeneous groups. However, LCA does not demand a predefined cut-off or a gold-standard measure as reference. For example, in traditional AP research, participants are considered AP possessors if they achieve an arbitrarily defined score (e.g., AP possessors must score 6 points or higher on an isolated pitch test). While previous research provides theoretical justifications for the cut-off choices, there is no statistical justification for choosing one cut-off threshold over another. Contrastingly, in LCA, class membership is inferred from the data and from the underlying patterns of responses across items. In this study, different numbers of classes were considered and evaluated.

The factor (IRT) mixture model was estimated based on Muthén [40], given that it is a generalization of the latent class model, where the assumption of conditional independence between the latent class indicators within a class is relaxed using a factor that influences the items within each class [40–42]. The factor represents individual variations in response probabilities within a class. Therefore, this model allows for heterogeneity within each class. As described in Mplus User's Guide (Example 7.27) [37], this model can be considered as an Item Response Theory (IRT) mixture model.

All three latent models were run twice, considering the two different rating criteria commonly used to define correct and incorrect answers. This choice was based on AP literature, as semitones errors can be considered incorrect [24, 43–45] or correct [31–33, 46] depending on how restrictively AP is defined. In our test, we adopted two criteria as follows:

1. **Perfect.** only exact pitch responses were considered correct; all other responses were incorrect (e.g., aural stimulus = *C*, correct response = *C*);

2. **Imperfect.** exact pitch responses and semitone deviations were considered correct; all other responses were incorrect (e.g., aural stimulus = *C*, correct response = *C*, *B*, or *C#/Db*).

The collected data formed a portrait of the latent trait distribution among the participants and was used to evaluate and validate the proposed test, i.e., how well it would measure the latent trait. The model fit indices used to evaluate and compare IRT and LCA were Akaike Information Criteria (AIC), Bayesian Information Criteria (BIC), and Simple Size Adjusted Bayesian Information Criterion (SSABIC). The lower the AIC, BIC, and SSABIC, the better the models being compared. In our case, we compared continuous versus categorical models under the same approach.

Due to the non-independence of sampling (i.e., students nested in universities), IRT and LCA models were run using robust maximum likelihood which produces standard errors, and chi-square test of the model fit considered this multilevel structure of the data [47, 48]. Lastly, a comparison between IRT and LCA models was conducted using BIC and AIC. Notably, given that there were three approaches to statistical modeling (i.e., IRT, LCA, and Hybrid modeling), comparisons were always made within the same criterion.

## Results

Ordinary descriptive statistics with the proportions and counts under both criteria ratings (*perfect* and *imperfect*) are shown in Table 1. The summing of correct answers for both rating

**Table 1. Frequency distribution.**

|  | *Perfect* (%) | *Imperfect* (%) |
|---|---|---|
| **Item a** | 6.9 | 40.5 |
| **Item b** | 39.2 | 46.4 |
| **Item c** | 11.4 | 31.2 |
| **Item d** | 28.2 | 34.0 |
| **Item e** | 43.4 | 53.1 |
| **Item f** | 42.1 | 47.6 |
| **Item g** | 27.6 | 37.0 |
| **Item h** | 29.1 | 49.4 |
| **Item i** | 7.7 | 34.4 |
| **Item j** | 9.6 | 47.0 |

This table provides the percentage of correct answers for p*erfect* and i*mperfect* approaches for each item.

criteria are shown in Table 2. It can be observed that the criterion for the *perfect* approach reduces the probability of a correct answer.

The results for IRT with two parameters (discrimination and difficulty) and three parameters (discrimination, difficulty, and guessing) for both criteria ratings (*perfect* and *imperfect*) are provided in Table 3.

The *perfect* approach under IRT with two parameters revealed item discrimination as moderate, high, and very high. The most discriminating item was item *g* (G6 on violin; 1.929) and the most difficult item was item *i* (G#1 on tuba; 2.419). The *imperfect* approach with two parameters showed item discrimination as low, moderate, and high. Item *f* displayed the highest item discrimination (C5 on piano; 1.697) and item *i* (G#1 on tuba; 0.926), identical to the *perfect* approach, showed the highest item difficulty.

For IRT with three parameters, results for the *perfect* approach showed item discriminations with high and very high values. The most discriminative item was item *b* (A5 on violin; 5.289) and the most difficult item, as in the IRT with two parameters, was item *i* (G#1 on tuba; 1.974). The guessing parameter demonstrated that item *b* had a high probability of being answered correctly (25.9%), even among those with very low ability to identify isolated pitches under the *perfect* approach. The *imperfect* approach with three parameters indicated item

**Table 2. Frequency distribution of a simple correct answers sum.**

| Sum of correct answers | *Perfect* (%) | *Imperfect* (%) |
|---|---|---|
| **0** | 16.9 | 2.9 |
| **1** | 26.2 | 9.6 |
| **2** | 19.3 | 18.9 |
| **3** | 13.5 | 15.5 |
| **4** | 8.3 | 15.8 |
| **5** | 3.4 | 10.9 |
| **6** | 5.0 | 6.3 |
| **7** | 3.1 | 4.7 |
| **8** | 2.2 | 5.4 |
| **9** | 1.0 | 5.2 |
| **10** | 1.1 | 4.9 |

This table provides the simple correct answers sum for each item for p*erfect* and i*mperfect* approaches.

**Table 3. Item response theory results: Two and three parameters for *perfect* and *imperfect* approaches.**

| | | IRT– 2 Parameters | | | | IRT– 3 Parameters | | | | | |
|---|---|---|---|---|---|---|---|---|---|---|---|
| **Perfect** | | **Discrimination** | **SE** | **Difficulty** | **SE** | **Discrimination** | **SE** | **Difficulty** | **SE** | **Guessing** | **SE** |
| | Item a | 1.927 | 0.197 | 2.05 | 0.267 | 2.942 | 0.555 | 1.886 | 0.237 | 0.018 | 0.008 |
| | Item b | 1.209 | 0.202 | 0.448 | 0.243 | 5.289 | 1.688 | 0.976 | 0.213 | 0.259 | 0.030 |
| | Item c | 1.391 | 0.295 | 1.937 | 0.434 | 3.484 | 0.482 | 1.664 | 0.240 | 0.048 | 0.008 |
| | Item d | 1.25 | 0.140 | 0.949 | 0.227 | 2.416 | 0.877 | 1.154 | 0.174 | 0.129 | 0.036 |
| | Item e | 1.129 | 0.164 | 0.277 | 0.190 | 1.490 | 0.371 | 0.547 | 0.348 | 0.117 | 0.102 |
| | Item f | 1.85 | 0.138 | 0.24 | 0.128 | 2.521 | 0.340 | 0.350 | 0.119 | 0.058 | 0.026 |
| | Item g | 1.929 | 0.391 | 0.767 | 0.205 | 2.999 | 0.640 | 0.889 | 0.179 | 0.070 | 0.028 |
| | Item h | 1.83 | 0.233 | 0.724 | 0.120 | 2.615 | 1.012 | 0.828 | 0.167 | 0.061 | 0.039 |
| | Item i | 1.303 | 0.239 | 2.419 | 0.261 | 3.021 | 0.489 | 1.974 | 0.096 | 0.034 | 0.010 |
| | Item j | 1.356 | 0.149 | 2.141 | 0.249 | 3.509 | 0.830 | 1.812 | 0.143 | 0.046 | 0.009 |
| **Imperfect** | Item a | 1.272 | 0.201 | 0.37 | 0.166 | 1.913 | 0.296 | 0.587 | 0.139 | 0.115 | 0.031 |
| | Item b | 1.189 | 0.219 | 0.131 | 0.212 | 3.669 | 1.308 | 0.836 | 0.258 | 0.309 | 0.062 |
| | Item c | 1.385 | 0.221 | 0.749 | 0.240 | 3.219 | 0.980 | 1.002 | 0.230 | 0.154 | 0.032 |
| | Item d | 1.293 | 0.135 | 0.652 | 0.203 | 2.421 | 0.719 | 0.897 | 0.167 | 0.142 | 0.039 |
| | Item e | 1.101 | 0.209 | -0.164 | 0.199 | 2.469 | 0.866 | 0.513 | 0.413 | 0.295 | 0.118 |
| | Item f | 1.697 | 0.244 | 0.045 | 0.145 | 3.612 | 2.368 | 0.458 | 0.235 | 0.210 | 0.118 |
| | Item g | 1.458 | 0.212 | 0.474 | 0.179 | 8.414 | 5.810 | 0.865 | 0.168 | 0.218 | 0.028 |
| | Item h | 1.209 | 0.193 | -0.001 | 0.186 | 8.401 | 5.824 | 0.832 | 0.144 | 0.363 | 0.038 |
| | Item i | 0.781 | 0.160 | 0.926 | 0.262 | 1.990 | 0.439 | 1.261 | 0.232 | 0.210 | 0.023 |
| | Item j | 0.614 | 0.177 | 0.203 | 0.107 | 1.137 | 0.332 | 0.911 | 0.377 | 0.240 | 0.089 |

This table provides the Item Response Theory results for each item for two and three parameters, in both the *perfect* and *imperfect* approaches. Item Response for two parameters shows discrimination and difficulty results. Item response for three parameters shows discrimination, difficulty, and guessing results. SE = Standard Error.

discrimination as moderate, high, and very high. Item *g* (G6 on violin; 8.414) demonstrated the highest discrimination parameter and item *i* (G#1 on tuba; 1.261) showed the highest item difficulty parameter. Under the *imperfect* approach, the probabilities of guessing increased across all the items. Item *h* indicated the highest guessing probability (36.3%), followed by item *b* (30.9%). Importantly, the standard errors (SE) were larger than under IRT with three parameters regardless of the adopted rating criterion, as commonly described in the literature [49, 50].

Table 4 shows the item-level fit. Under IRT with two parameters, the *imperfect* rating indicated that all items had a good fit (S-$X^2$ $p > 0.05$). However, under the *perfect* approach, two out of the ten items (items *c* and *d*) were statistically significant. Under IRT with three parameters, the majority of the items displayed a reduction in *p*-values when compared to two parameters, for both approaches.

A reason for the *imperfect* approach under two parameters having a better item level fit may be due to the increase in the probabilities of answering the items correctly (i.e., proportion and counts were higher since it was a less strict criterion). For items *c* and *d*–scored with the criterion of *perfect* rating–misfit is illustrated by comparisons of predicted and observed proportion of correct results (Figs 2 and 3). In particular, higher than expected proportion of correct answers are seen for theta scores a little higher than 1 and for theta scores a little lower than -1

Table 5 depicts the model fit for IRT models (two and three items parameters) for the *perfect* and *imperfect* ratings. Considering the *perfect* approach, the lowest BIC was in favor of an IRT model with two parameters. However, for the *imperfect* approach, the lowest BIC was in favor of an IRT model with three parameters. Notably, for *perfect* scoring, evaluations of item

**Table 4. Item level fit for *perfect* and *imperfect* ratings.**

| | | | 2 Parameters | | | | 3 Parameters | | | |
|---|---|---|---|---|---|---|---|---|---|---|
| | | | S-X2 | df(S-X2) | RMSEA | p-value | S-X2 | df(S-X2) | RMSEA | p-value |
| *Perfect* | **Continuous** | **Item a** | 7.448 | 7 | 0.009 | 0.384 | 4.651 | 6 | <0.001 | 0.589 |
| | | **Item b** | 11.039 | 6 | 0.033 | 0.087 | 4.946 | 3 | 0.029 | 0.176 |
| | | **Item c** | 18.967 | 7 | 0.047 | 0.008 | 13.763 | 5 | 0.047 | 0.017 |
| | | **Item d** | 14.791 | 6 | 0.043 | 0.022 | 15.843 | 5 | 0.053 | 0.007 |
| | | **Item e** | 7.582 | 6 | 0.018 | 0.27 | 4.322 | 4 | 0.010 | 0.364 |
| | | **Item f** | 5.037 | 5 | 0.003 | 0.411 | 7.979 | 4 | 0.036 | 0.092 |
| | | **Item g** | 1.466 | 5 | <0.001 | 0.917 | 1.58 | 4 | <0.001 | 0.812 |
| | | **Item h** | 1.225 | 5 | <0.001 | 0.942 | 1.444 | 4 | <0.001 | 0.837 |
| | | **Item i** | 7.017 | 7 | 0.002 | 0.427 | 4.891 | 6 | <0.001 | 0.558 |
| | | **Item j** | 11.588 | 7 | 0.029 | 0.115 | 11.044 | 6 | 0.033 | 0.087 |
| *Imperfect* | | **Item a** | 11.949 | 7 | 0.03 | 0.102 | 13.602 | 6 | 0.040 | 0.034 |
| | | **Item b** | 5.057 | 7 | <0.001 | 0.653 | 4.82 | 6 | <0.001 | 0.567 |
| | | **Item c** | 9.43 | 7 | 0.021 | 0.223 | 7.738 | 6 | 0.019 | 0.258 |
| | | **Item d** | 5.828 | 7 | <0.001 | 0.560 | 6.707 | 6 | 0.012 | 0.349 |
| | | **Item e** | 8.401 | 7 | 0.016 | 0.299 | 7.975 | 6 | 0.021 | 0.24 |
| | | **Item f** | 5.24 | 7 | <0.001 | 0.631 | 8.738 | 5 | 0.031 | 0.12 |
| | | **Item g** | 7.96 | 7 | 0.013 | 0.336 | 6.334 | 5 | 0.018 | 0.275 |
| | | **Item h** | 6.738 | 7 | <0.001 | 0.457 | 15.606 | 5 | 0.052 | 0.008 |
| | | **Item i** | 3.038 | 7 | <0.001 | 0.881 | 3.978 | 6 | <0.001 | 0.68 |
| | | **Item j** | 6.991 | 7 | <0.001 | 0.430 | 5.039 | 6 | <0.001 | 0.539 |

This table provides the item-level fit values for each item for two and three parameters, in both *perfect* and *imperfect* approaches. S-X2 is an item fit index for dichotomous item response theory models. df(S-X2) is the degree of freedom for item fit index for dichotomous item response theory models. RMSEA = (Root Mean Square Error of Approximation).

fit showed significant misfits for items *c* and *d*. This suggests that these two items are problematic as indicators of the latent trait. Moreover, for the *imperfect* approach, the standard errors of the discrimination parameters were high. Therefore, for both *perfect* and *imperfect* models, we concluded that the two-parameter model fits better than three-parameter model.

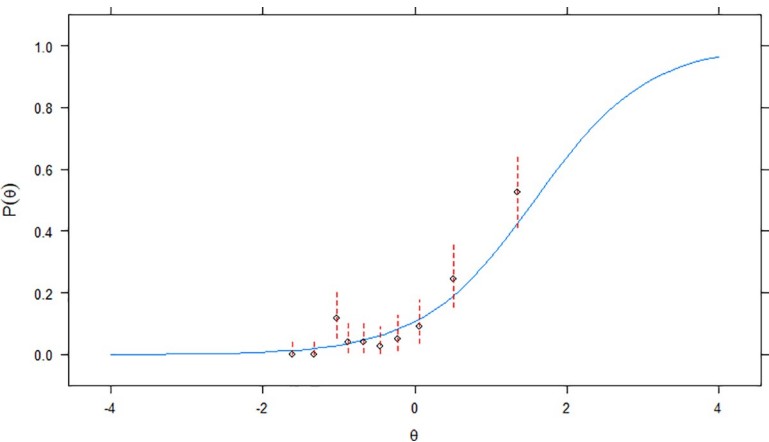

**Fig 2. Empirical plots (item *c*) for *perfect* model with 2 parameters.** Confidence intervals for the probability of endorsement of item *c*, correctly given the amount of AIPWR, are represented in dashed red lines. The estimated item characteristic curve for item *c* is indicated in continuous blue lines.

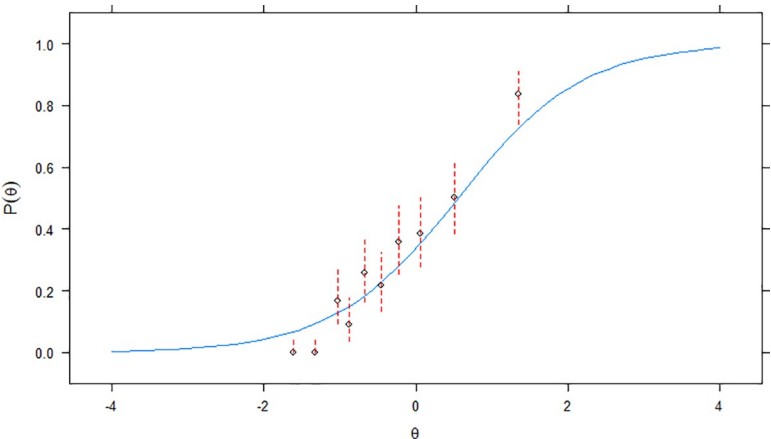

**Fig 3. Empirical plots (item *d*) for *perfect* model with 2 parameters.** Confidence intervals for the probability of endorsement of item *d*, correctly given the amount of AIPWR, are represented in dashed red lines. The estimated item characteristic curve for item *d* is indicated in continuous blue lines.

LCA results indicate the best solution for the two classes for both *perfect* and *imperfect* approaches, as illustrated in Table 6.

The best class solution was two classes, given the strongest decline in the AIC and BIC values. There was still a reduction from the two to three-classes solution in BIC and AIC values, which was expected. However, such information gain is insufficient for the justification of an additional extracted class when compared to the information gain (i.e., the reduction of BIC and AIC) from one to two classes. Figs 4 and 5 show the LCA results for *perfect* and *imperfect* LCA results.

The figures illustrate that one group had higher a probability of correctly identifying the pitches (depicted by the red line, representing 16.3% of the sample for *perfect* approach and 20.9% of the sample for *imperfect* approach). Contrarily, the other group had a lower probability of correctly identifying the stimuli (blue line, 83.7% for *perfect* approach and 79.1% for *imperfect* approach). Notably, even the red group did not achieve a value of 1 for any of the items, which would indicate a 100% probability of answering correctly for a giving stimulus. Moreover, the prevalence of the group with the highest probabilities of correctly identifying the pitches was lower than the group with lowest probabilities of correctly identifying the pitches.

**Table 5. Model fit information for IRT models—*perfect* and *imperfect*, two and three parameters.**

| | | | Number of Classes | Free Parameters | Loglikelihood Correction Factor for MLR | Loglikelihood (HO value) | Akaike (AIC) | Bayesian (BIC) | SSA (BIC) |
|---|---|---|---|---|---|---|---|---|---|
| *Perfect* | Continuous | 2 Par. | —— | 20 | 1.6634 | -3474.444 | 6988.887 | 7082.150 | 7018.640 |
| | | 3 Par. | —— | 30 | 1.3837 | -3449.099 | 6958.198 | 7098.092 | 7002.827 |
| *Imperfect* | | 2 Par. | —— | 20 | 2.1810 | -4816.900 | 9673.801 | 9767.064 | 9703.554 |
| | | 3 Par. | —— | 30 | 1.6942 | -4766.699 | 9593.398 | 9733.291 | 9638.026 |

This table provides the model fit information for two and three parameters, in both *perfect* and *imperfect* approaches. MLR (Maximum Likelihood Robust). AIC (Consistent Akaike's Information Criterion). BIC (Bayesian Information Criterion). SSA (BIC) (Simple Size Adjusted Bayesian Information Criterion).

**Table 6. Latent class analysis results for *perfect* and *imperfect* approaches.**

| | | Number of Classes | Free Parameters | Loglikelihood Correction Factor for MLR | Loglikelihood (HO value) | Akaike (AIC) | Bayesian (BIC) | SSA (BIC) | VLMR LRT (p-value) | LMR LR adjusted test | Entropy |
|---|---|---|---|---|---|---|---|---|---|---|---|
| *Perfect* | Categorical | 1 | 10 | 4.2098 | -3925.508 | 7871.016 | 7917.648 | 7885.893 | ——— | ——— | ——— |
| | | 2 | 21 | 1.8203 | -3482.636 | 7007.273 | 7105.199 | 7038.513 | 0.0212 | 0.0219 | 0.914 |
| | | 3 | 32 | 1.5176 | -3444.222 | 6952.444 | 7101.664 | 7000.048 | 0.5114 | 0.5130 | 0.656 |
| | | 4 | 43 | 1.4185 | -3423.966 | 6933.931 | 7134.446 | 6997.899 | 0.5677 | 0.5684 | 0.678 |
| | | 5 | 54 | 1.2412 | -3407.555 | 6923.11 | 7174.919 | 7003.442 | 0.4955 | 0.4960 | 0.675 |
| | | 6 | 65 | 1.1888 | -3392.677 | 6915.335 | 7218.438 | 7012.031 | 0.4577 | 0.4581 | 0.718 |
| | | 7 | 76 | 1.2128 | -3384.837 | 6921.745 | 7276.143 | 7034.805 | 0.6390 | 0.6393 | 0.732 |
| *Imperfect* | | 1 | 10 | 4.6859 | -5243.614 | 10507.229 | 10553.86 | 10522.105 | ——— | ——— | ——— |
| | | 2 | 21 | 2.0328 | -4784.795 | 9611.589 | 9709.515 | 9642.829 | 0.0160 | 0.0165 | 0.893 |
| | | 3 | 32 | 1.6635 | -4755.888 | 9575.775 | 9724.995 | 9623.379 | 0.5189 | 0.5200 | 0.688 |
| | | 4 | 43 | 1.6857 | -4736.214 | 9558.428 | 9758.943 | 9622.396 | 0.7612 | 0.7614 | 0.768 |
| | | 5 | 54 | 1.4823 | -4716.346 | 9540.692 | 9792.501 | 9621.024 | 0.4518 | 0.4523 | 0.797 |
| | | 6 | 65 | 1.4254 | -4708.111 | 9546.223 | 9849.326 | 9642.919 | 0.6568 | 0.6576 | 0.651 |
| | | 7 | 76 | 1.3179 | -4697.236 | 9546.471 | 9900.869 | 9659.531 | 0.4955 | 0.4951 | 0.709 |
| | | 8 | 87 | 1.2404 | -4690.011 | 9554.022 | 9959.714 | 9683.445 | 0.5387 | 0.5393 | 0.742 |
| | | 9 | 98 | 1.2126 | -4685.470 | 9566.94 | 10023.927 | 9712.728 | 0.5037 | 0.5038 | 0.685 |

This table provides the Latent Class Analysis for both *perfect* (7 classes) and *imperfect* (9 classes) approaches. MLR (Maximum Likelihood Robust). AIC (Consistent Akaike's Information Criterion). BIC (Bayesian Information Criterion). SSA (BIC) (Simple Size Adjusted Bayesian Information Criterion). VLMR LRT (Vuong-LO-Mendell-Rubin Likelihood Ratio Test). LMR (Likelihood Mendell Rubin).

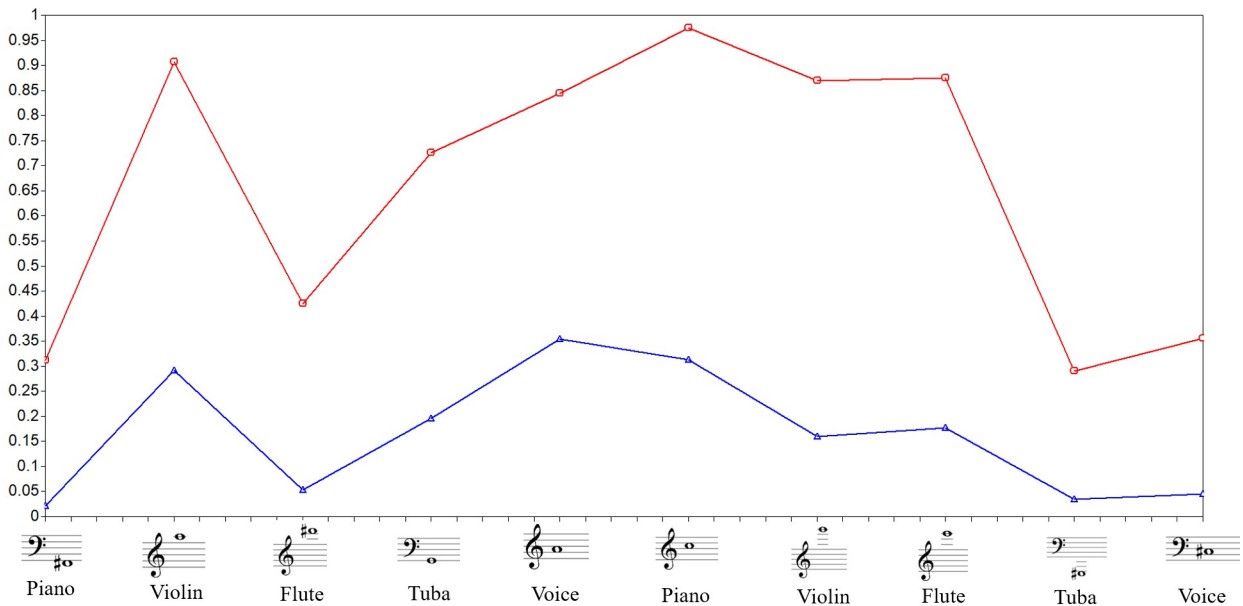

**Fig 4. Isolated pitch *perfect* approach—latent class analysis for two classes.** Class 1 (red line—16.3%) represents the population with greater ability to identify pitches in various registers and timbres, without reference. Class 2 (blue line—83.7%) represents the population with less ability to identify pitches in various registers and timbres, without reference. The y-axis represents the probability of a correct answer and the x-axis represents each item tested. Figure adapted from Germano et al. [8].

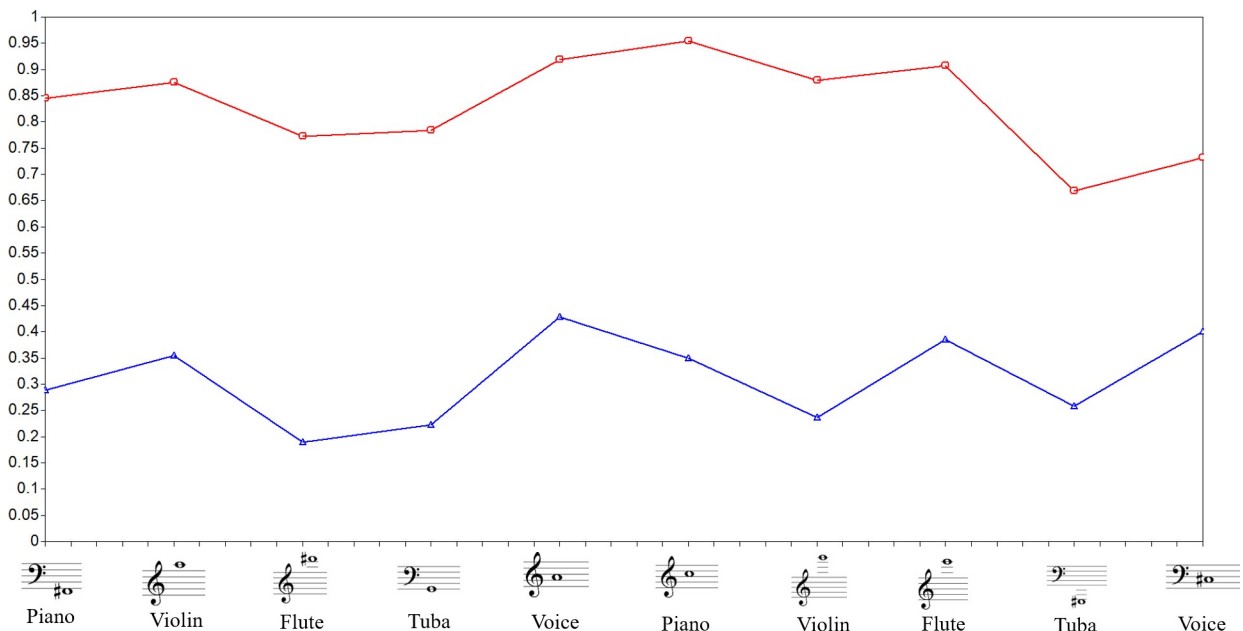

**Fig 5. Isolated pitch *imperfect* approach—latent class analysis for two classes.** Class 1 (red line—20.9%) represents the population with greater ability to identify pitches with semitone deviations in various registers and timbres without reference. Class 2 (blue line—79.1%) represents the population with less ability to identify pitches with semitone deviations in variated registers and timbres without reference. The y-axis represents the probability of a correct answer and the x-axis represents each item tested.

Based on the results from the continuous and categorical options, the hybrid model was conducted by merging the features of the best solutions obtained from both modeling approaches, the two classes solution, and a unidimensional solution.

The hybrid model fit information is given in Table 7.

Based on model fit information, we conclude that the continuous solution was the best solution for the *perfect* approach, with lower BIC than the categorical and hybrid solutions. This indicates that the ability to recognize isolated pitches in different timbres and registers without reference is better modeled as a continuous ability, rather than when the *perfect* rating approach is considered with either a categorical or a hybrid model. Alternatively, the categorical solution demonstrated the best solution for the *imperfect* approach, with lower BIC than the continuous and hybrid solutions. This indicates that adopting flexibility in isolated pitch recognition without reference (with semitone deviations considered as correct) is better modeled as latent groups.

Fig 6 shows a histogram of the continuous distribution of the ability to recognize isolated pitches under the *perfect* approach, and Fig 7 displays the *imperfect* approach. The *perfect*

**Table 7. Hybrid model for *perfect* and *imperfect* ratings.**

| | | Number of Classes | Free Parameters | Loglikelihood Correction Factor for MLR | Loglikelihood (HO value) | Akaike (AIC) | Bayesian (BIC) | SSA (BIC) | Entropy |
|---|---|---|---|---|---|---|---|---|---|
| *Perfect* | Hybrid | 2 | 43 | 1.2784 | -3418.603 | 6923.206 | 7123.721 | 6987.174 | 0.876 |
| *Imperfect* | | 2 | 43 | 2.8625 | -4736.865 | 9559.731 | 9760.245 | 9623.699 | 0.759 |

This table provides the hybrid model for *perfect* and *imperfect* ratings with two latent classes and a unidimensional underlying latent factor. MLR (Maximum Likelihood Robust). AIC (Consistent Akaike's Information Criterion). BIC (Bayesian Information Criterion). SSA (BIC) (Simple Size Adjusted Bayesian Information Criterion).

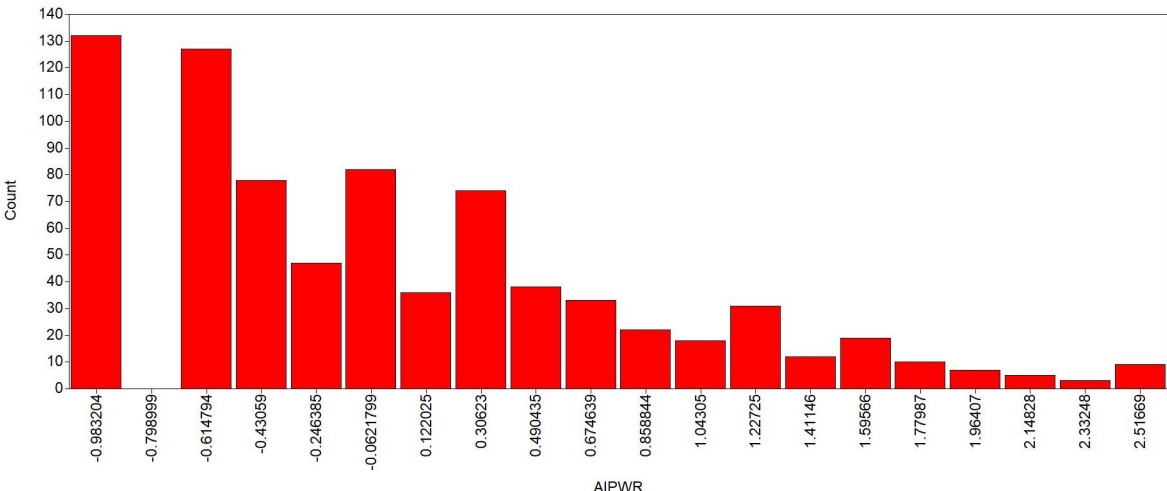

**Fig 6. Isolated pitch *perfect* approach—histograms (sample values, estimated factor scores, estimated values, residuals).** *Perfect* approach ability. AIPWR: Ability to identify Isolated Pitch Without Reference. The y-axis represents the number of individuals. The x-axis represents the ability divided into 20 columns.

approach (Fig 6) displays a half-normal distribution, while the *imperfect* approach (Fig 7) displays a log-Cauchy like distribution.

## Discussion

Our results demonstrate a good fit adjustment in measuring the ability to recognize isolated pitches without reference in a continuous solution of *perfect* rating criteria. When the *imperfect* approach is used as a rating criterion, a categorical solution is preferred.

Moreover, in a two-parameter IRT model for the *perfect* scoring approach, all the items showed high values of discrimination. This indicates that our set of stimuli were appropriate for discriminating between subjects with high and low abilities to recognize isolated pitches.

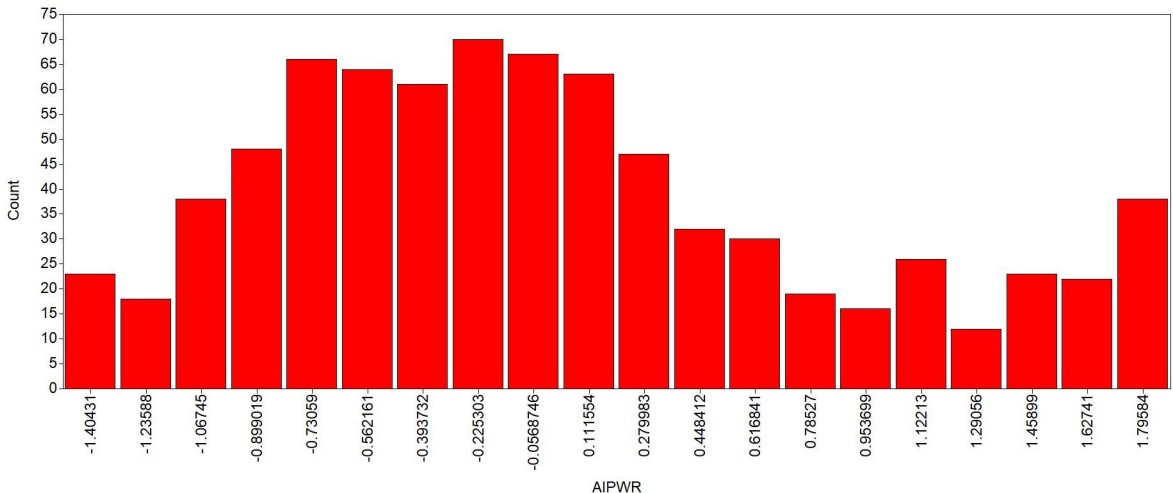

**Fig 7. Isolated pitch *imperfect* approach—histograms (sample values, estimated factor scores, estimated values, residuals).** *Imperfect* approach ability. AIPWR: Ability to identify Isolated Pitch Without Reference. The y-axis represents the number of individuals. The x-axis represents the ability divided into 20 columns.

The items' difficulty values in both the *perfect* and *imperfect* approaches were high (with the exception of items *e* and *h* in the *imperfect* approach with two parameters). This was an expected result, as the identification of pitches without any reference is considered to be an exceedingly challenging task for most musicians. Notably, we could not formally compare the *imperfect* and *perfect* approaches regarding superiority, because they are not *nested models* [51].

When comparing LCA to IRT, our results indicated that the ability to recognize isolated pitches was better represented by a continuous model for the *perfect* approach. That is, through a continuous line where participants were arranged according to their degree of ability, as can be seen in (Fig 6). This is a highly unexpected result, because a consensually adopted methodology in AP research is the division of subjects into two categories. Here, we labeled both groups as high-skilled and low-skilled.

Alternatively, the common division adopted by most AP research (dividing the population in two groups) is the best solution only when using the *imperfect* approach as a rating criterion. Crucially, our results demonstrate how the two rating approaches commonly used in AP literature (*perfect* and *imperfect*) might influence the decision of the best model underlying isolated pitch recognition ability.

In theory, it was expected that both the *perfect* and *imperfect* approaches would be better represented by a categorical model, because this is a status quo in the field of AP. However, the *perfect* approach showed the continuous model as the best solution. This was greatly unexpected, as the *perfect* approach uses more restrictive criteria than the *imperfect* approach does. According to the literature, AP possessors make many semitone errors. We thus hypothesize that these restrictions allow us to capture more fine grain variations of IWRPV skills across the participants sampling. Using the *perfect* scoring approach, 1.1% of participants had all items correct. According to the IRT model, these participants would be expected to have greater skills in isolated pitch recognition tasks than participants with lower numbers of correct responses. In contrast, for the *imperfect* scoring approach, the LCA model assumes that 20.9% of participants have high skills in isolated pitch recognition tasks. Within this group further differentiation in skills cannot be made. The 4.9% who had all 10 responses correct using the *imperfect* scoring approach were just luckier than the remaining 16% in the high-skill group. More research is necessary to examine the causes for the differences in the underlying models.

It is especially important to understand that a high ability to identify pitches without reference (as a latent group) is not necessarily synonymous with being an AP possessor. Furthermore, a low skill is not necessarily synonymous with not being an AP possessor. This is because we cannot deduce that a high performance in isolated pitch recognition is due to the presence of AP ability, since well-trained musicians that are non-AP possessors can also possibly have a high performance. This kind of test is not capable of assessing whether a participant is automatically associating a pitch to a verbal label.

In many areas, it is common procedure to choose a cut-off threshold to categorize subjects into a certain group, even when the original measure is continuous. Results from LCA may be exported from Mplus (or other statistical packages dealing with mixture modeling). This generates a most likely class membership and each subject would have a conditional probability for each group. That is, the probability of being classified as likely to correctly answer and the probability of being classified as less likely to correctly answer.

Interestingly, we observed that none of the individual stimuli were answered correctly 100% of the time, even among the group classified as showing a high probability of choosing the correct answer (less than 20% of the 783 participants). These incorrect rates among those classified as having higher probabilities of performing well in isolated pitch discrimination tasks corroborates previous research indicating that participants are fallible and can make a

considerable number of mistakes. In terms of limitation, future studies may investigate more detailed elements of psychometrics as local dependency for each of the models (IRT and LCA), invariance testing per sex, time of studies, and played instruments.

## Conclusion

The latent approach elucidates the psychometrics features for the measurement of isolated pitch recognition ability in a large-scale evaluation, which can be adopted by future researchers. According to model fit information indices, the test measures the proposed latent trait of AIPWR ability very well, given that the stimuli varied according to difficulty and discriminatory levels. The *perfect* approach showed a better adjustment through a continuous line and the *imperfect* approach showed a better adjustment when dividing the population in two groups. It is important to note that the ten stimuli did not evaluate whether a participant made an automatic association between a certain pitch and a learned verbal label. Consequently, we could not conclude that a high score in our test indicates that the participant possesses AP or that a low score indicates that they do not. The only plausible conclusion is higher scores indicate more latent trait in the participant, while a lower score indicates less latent trait. These findings may contribute to a better theoretical understanding of AP ability, showing that different rating criteria in AP tests greatly influence test results and the measurement of AP ability.

## Supporting information

**S1 Data.**
(XLSX)

**S1 File.**
(DOCX)

## Acknowledgments

We thank all students who volunteered in this research, as well as the professors and universities for allowing the conducting of this test.

## Author Contributions

**Conceptualization:** Nayana Di Giuseppe Germano, Hugo Cogo-Moreira, Graziela Bortz.

**Data curation:** Nayana Di Giuseppe Germano, Hugo Cogo-Moreira.

**Formal analysis:** Nayana Di Giuseppe Germano, Hugo Cogo-Moreira, Fausto Coutinho-Lourenço.

**Investigation:** Nayana Di Giuseppe Germano, Hugo Cogo-Moreira, Graziela Bortz.

**Methodology:** Nayana Di Giuseppe Germano, Hugo Cogo-Moreira, Graziela Bortz.

**Supervision:** Hugo Cogo-Moreira, Graziela Bortz.

**Validation:** Nayana Di Giuseppe Germano, Hugo Cogo-Moreira.

**Visualization:** Nayana Di Giuseppe Germano.

**Writing – original draft:** Nayana Di Giuseppe Germano, Hugo Cogo-Moreira, Graziela Bortz.

**Writing – review & editing:** Nayana Di Giuseppe Germano, Hugo Cogo-Moreira, Fausto Coutinho-Lourenço, Graziela Bortz.

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
