## [Decision Letter · Decision Letter 0]

1 May 2020

PONE-D-20-04410

Absolute Pitch as a Latent Trait

PLOS ONE

Dear Dra. Germano,

Thank you for submitting your manuscript to PLOS ONE. After careful consideration, we feel that it has merit but does not fully meet PLOS ONE’s publication criteria as it currently stands. Therefore, we invite you to submit a revised version of the manuscript that addresses the points raised during the review process.

I found the manuscript interesting in that it tries to evaluate if absolute pitch is a trait or a categorical variable. The choice of methodology is appropriate, but the reporting is lacking in quality. The comments from the two reviewers are very constructive and should enable you to improve the manuscript. 

We would appreciate receiving your revised manuscript by Jun 15 2020 11:59PM. To enhance the reproducibility of your results, we recommend that if applicable you deposit your laboratory protocols in protocols.io, where a protocol can be assigned its own identifier (DOI) such that it can be cited independently in the future. For instructions see: http://journals.plos.org/plosone/s/submission-guidelines#loc-laboratory-protocols

We look forward to receiving your revised manuscript.

Kind regards,

Karl Bang Christensen, Ph.D.

Academic Editor

PLOS ONE

Additional Editor Comments:

Please provide a revised version addressing the comments from the two reviewers, who are very constructive and have provided comments that should enable you to improve the manuscript. The choice of methodology is appropriate, but the reporting is lacking in quality. You must address test of validity much more rigorously in a revised version. Evaluate fit of a one-dimensional CFA model (reporting chi-square, df and P-value). If you also waht to report the RMSEA with corresponding confidence interval or other indeces of close fit that is OK.

The figures are attached on their own with no legends or titles. Some of them are not even mentioned in the main text of the paper. One example line 145 states 'fig. 1', but the next line appears to discuss fig. 2?

Journal Requirements:

1. Please ensure that your manuscript meets PLOS ONE's style requirements, including those for file naming. The PLOS ONE style templates can be found at:

2. Please modify the title to ensure that it is meeting PLOS’ guidelines (https://journals.plos.org/plosone/s/submission-guidelines#loc-title).

In particular, the title should be "specific, descriptive, concise, and comprehensible to readers outside the field" and in this case it is not informative and specific about your study's scope and methodology.  When modifying the title please be sure to amend both the title on the online submission form (via Edit Submission) and the title in the manuscript so that they are identical.

3. Your ethics statement must appear in the Methods section of your manuscript. If your ethics statement is written in any section besides the Methods, please move it to the Methods section and delete it from any other section. Please also ensure that your ethics statement is included in your manuscript, as the ethics section of your online submission will not be published alongside your manuscript.

Reviewers' comments:

Reviewer's Responses to Questions

**Comments to the Author**

1. Is the manuscript technically sound, and do the data support the conclusions?

Reviewer #1: Partly

Reviewer #2: No

2. Has the statistical analysis been performed appropriately and rigorously? 

Reviewer #1: No

Reviewer #2: No

3. Have the authors made all data underlying the findings in their manuscript fully available?

Reviewer #1: No

Reviewer #2: No

4. Is the manuscript presented in an intelligible fashion and written in standard English?

Reviewer #1: Yes

Reviewer #2: No

5. Review Comments to the Author

Reviewer #1: PONE-D-20-04410

This is an interesting application of modern psychometric methods. The comparison of IRT models and latent class models appear well suited for the theoretical problem posed. I have some suggestions for the data analysis and presentation of results.

1. You should present some basic descriptive information, e.g. the frequency distribution for each item (e.g. Perfect, Imperfect, Wrong), and the frequency distribution of a simple sum of the items (or two sums according to your two approaches). This allows the reader to get some sense of your data.

2. You estimate a 2-parameter and a 3-parameter IRT model. However, it is unclear which of these models you deem is having the best fit. Also, it was unclear to me, which of these models you compare with the latent class model and for which IRT model you present the score distribution in figure 5. Please clarify.

3. Fit is evaluated through a global chi-square test. However, chi-square tests with app. 1000 DF are not optimal. You report χ2(991) = 6635.882, p-value =0.999 (line 239). This must be a typo, if the chisq value is correct, p=0. I suggest you do two things.

3a. For global tests of fit and comparisons of the 2-P and 3-P models, use AIC and BIC as in your comparison with latent class models.

3b. Evaluate item level fit, e.g. by the item fit tests suggested by Orlando and Thissen (Applied Psychological Measurement 2000). Such tests are available in the IRTPRO software and the free R package mirt (https://cran.r-project.org/web/packages/mirt/mirt.pdf). Such item based fit test may identify some stimuli that are not well modeled by IRT.

4. Your IRT parameter estimates have very large standard errors, in particular for the discrimination parameter in the 3-P model. Difficulties of estimating the discrimination parameter in 3-P models is a known problem, but I am still concerned. You may want to use a prior for the discrimination parameter in addition to a prior for the guessing parameter.

5. I am also a bit worried about the magnitude for the guessing parameter for some items. For pure guessing, you would expect a guessing parameter around 1/12 = 0.08. Is the any theoretical that some items would have a lower asymptote of 0.3?

6. In choosing between latent class model, you argue that the improved fit of the models with more than 2 classes should be ignored, due to the complexity of these models. However, in comparisons between IRT and latent class models for the perfect approach, you suddenly argue that a difference in fit of the same magnitude is important and should be interpreted. You cannot have it both ways. If you compare e.g. a 3 class model with the IRT model, you get the following results:

AIC D AIC 3CL BIC D BIC 3CL SSABIC D SSABIC 3CL

Perfect 6988.887 6952.444 7082.150 7101.664 7018.640 7000.048

This comparison show no particular superiority of the IRT model in terms of fit. The same reasoning could be applied for a 4-CL or a 5-CL model. You may want to keep a 2-CL model in the comparisons for conceptual reasons, but you should include a latent class model with better fit (e.g. 3 classes or 4 classes). Based on the results I have seen, I would conclude that for the “perfect” approach, a latent class model with 3, 4, or 5 classes has equally good fit as an IRT model. For the “imperfect” approach, the IRT model seems clearly better.

7. You present the score distribution for one IRT model, presumably for the “perfect” approach (the text in lines 272-273 seems to have the numbering of figures wrong). The score distribution is clearly skewed. A large group of people seem to have the same level of ability to identify the pitch of a tone (i.e. no ability). This may pose a problem for standard IRT model estimation, since a normal distribution is assumed for the latent trait. For this reason, a better model for your data may be a latent mixture distribution model with 2 latent classes. Class one consist of persons who are not able to identify the pitch regardless of the stimulus. Class two consists of persons with at least some ability to identify pitch. Within class two responses might follow a 2-P IRT model. Such a model can be estimated in Mplus. It is fairly complex, but you may want to at least discuss it.

8. With regards to the psychometric lingo, I would suggest that :

8a. “Psychometric” is better than “Psychometrical”

8b. “Latent variable models” is better than “structural equation model”. SEM refers to a particular type of latent variable models, a type you do not use in your analysis.

8c. “Continuous” is better than “Dimensional”

Specific suggestions:

Line 24. I suggest “Through Latent Variable Models (LVM) we can evaluate consistency validity…”

Line 27. I suggest “… two LVM approaches: continuous latent variables (LV)…”

Line 37, I suggest “The phenomenon of absolute pitch (AP)…”

Line 65. It might be helpful to define relative pitch.

Line 115. Is there a reference regarding the use of a combination of AP and RP to identify pitches?

Line 159. I suggest “…continuous and categorical.”

Line 160. I suggest writing “The former approach, Item Response Theory (IRT), …”

Lines 166-170. I suggest writing: “a) An IRT model with two parameters for each stimuli: the discrimination parameter (also called parameter a), which describe the ability of this stimuli to distinguish between persons with low and high pitch identification ability and the item location parameter (also called parameter b), representing the level of pitch identification ability where you have 50% chance of correctly identifying the pitch of this stimulus.”

Lines 172+173. I suggest writing: “… is the probability of a person with very low pitch identification ability still correctly get a correct answer for a given stimulus.

Reviewer #2: The ultimate aim of this research project is not explicitly stated, and it remains a little unclear. The authors have created a new test of assessing absolute pitch, and this was investigated using an item-response theory (IRT) approach and a latent class analysis (LCA) approach. These models were then compared to see which offered the best fit.

Although the foundation and rationale for the study seems reasonable, in that the authors wish to create a measure to determine whether individuals have the ability to identify absolute pitch, the applied methodologies are confusing and it should be better explained as to why they are appropriate.

• Latent Class Analysis categorises people into groups, under the assumption that the same thing is being measured for all people, by a standardised count, process, or measurement device/scale.

• IRT is used to determine whether a set of items are delivering a valid total score of an unobservable latent trait.

Thus it should be explained in more detail why the fit of these models should be compared as they look at different things. It should be emphasized that the categorisation of people relies on the measurement process being valid and stable, so in some sense LCA should not be considered until the measure has been validated.

For the IRT scale assessment approach, there are also many aspects that have been neglected.

There is no indication of item fit. No investigation of response dependency. No assessment of reliability or targeting.

Was a single parameter model considered? - A single-parameter (Rasch) model would be appropriate for scale development and validation purposes, and for determining whether a total score from a set of items is a sufficient statistic to assess the level of a latent trait.

Additionally, for the ‘imperfect approach’, it may be worth the authors considering a partial-credit model, where an exact pitch classification is awarded a score of 2, a semitone deviation is awarded 1, and all other pitches are scored 0.

There are also some further issues within the manuscript that would need attention:

The model fit statistics are dubious, and there is no real interpretation of the fit statistics that are presented. Certainly a test with 1000 degrees of freedom will have no statistical power.

In the manuscript, it is stated that items are centred around the 0 location, but there are no item locations reported below 0 – where are they centred?

The pitch test is based across different musical instruments – have these instruments been calibrated? Has the pitch been externally verified in some way?

Additionally, the manuscript is currently in need of a language edit and the Figures are incorrectly labelled.

6. PLOS authors have the option to publish the peer review history of their article (what does this mean?). If published, this will include your full peer review and any attached files.

Reviewer #1: Yes: Jakob Bue Bjorner

Reviewer #2: No

---

## [Author Response · Author response to Decision Letter 0]

14 Jul 2020

PONE-D-20-04410

Absolute Pitch as a Latent Trait

PLOS ONE

Dear Dra. Germano,

Thank you for submitting your manuscript to PLOS ONE. After careful consideration, we feel that it has merit but does not fully meet PLOS ONE’s publication criteria as it currently stands. Therefore, we invite you to submit a revised version of the manuscript that addresses the points raised during the review process.

I found the manuscript interesting in that it tries to evaluate if absolute pitch is a trait or a categorical variable. The choice of methodology is appropriate, but the reporting is lacking in quality. The comments from the two reviewers are very constructive and should enable you to improve the manuscript. 

Answer: Dear Editor, we are thankful for the opportunity to answer the insightful comments we received. We also would like to state that for solving one of the issues, we needed extra help, which was provided by a newly added co-author, Fausto Lourenco Coutinho. He was fundamental to answer and deal with some issues involving R.

We would appreciate receiving your revised manuscript by Jun 15 2020 11:59PM. To enhance the reproducibility of your results, we recommend that if applicable you deposit your laboratory protocols in protocols.io, where a protocol can be assigned its own identifier (DOI) such that it can be cited independently in the future. For instructions see: http://journals.plos.org/plosone/s/submission-guidelines#loc-laboratory-protocols

• A rebuttal letter that responds to each point raised by the academic editor and reviewer(s). This letter should be uploaded as separate file and labeled 'Response to Reviewers'.

• A marked-up copy of your manuscript that highlights changes made to the original version. This file should be uploaded as separate file and labeled 'Revised Manuscript with Track Changes'.

• An unmarked version of your revised paper without tracked changes. This file should be uploaded as separate file and labeled 'Manuscript'.

We look forward to receiving your revised manuscript.

Kind regards,

Karl Bang Christensen, Ph.D.

Academic Editor

PLOS ONE

Additional Editor Comments:

Please provide a revised version addressing the comments from the two reviewers, who are very constructive and have provided comments that should enable you to improve the manuscript. The choice of methodology is appropriate, but the reporting is lacking in quality. You must address test of validity much more rigorously in a revised version. Evaluate fit of a one-dimensional CFA model (reporting chi-square, df and P-value). If you also waht to report the RMSEA with corresponding confidence interval or other indeces of close fit that is OK.

Answer: We thank you for the opportunity to answer all the issues raised by the referees. An important detail regarding our IRT models: differently of Mplus default estimator operating under WLSMV (probit link function), which generates CFI, TLI, RMSEA, we are using a different estimator called robust Maximum-likelihood estimator, which does not generate, under dichotomous items, those above cited global model fit indices. To improve the quality of our report, we did extra data analysis (as suggested by the Reviewer#1) and the results are detailed in the following answers. In order to conduct the extra analysis and interpretation, because they involved R, we invited Fausto Lourenco Coutinho, who is expert in R and a PHD student supervised by Hugo Cogo-Moreira, to join us in this manuscript. 

The figures are attached on their own with no legends or titles. Some of them are not even mentioned in the main text of the paper. One example line 145 states 'fig. 1', but the next line appears to discuss fig. 2?

Answer: Legends and better descriptions regarding the tables were provided. 

Journal Requirements: When submitting your revision, we need you to address these additional requirements.

1. Please ensure that your manuscript meets PLOS ONE's style requirements, including those for file naming. The PLOS ONE style templates can be found at:

Answer: The manuscript was adjusted in order to meet PLOS ONE’s style requirements. We appreciated the links provided.

2. Please modify the title to ensure that it is meeting PLOS’ guidelines (https://journals.plos.org/plosone/s/submission-guidelines#loc-title).

In particular, the title should be "specific, descriptive, concise, and comprehensible to readers outside the field" and in this case it is not informative and specific about your study's scope and methodology. When modifying the title please be sure to amend both the title on the online submission form (via Edit Submission) and the title in the manuscript so that they are identical.

Answer: We changed the title to a more specific and descriptive version, which should be more comprehensible to readers outside the field. 

3. Your ethics statement must appear in the Methods section of your manuscript. If your ethics statement is written in any section besides the Methods, please move it to the Methods section and delete it from any other section. Please also ensure that your ethics statement is included in your manuscript, as the ethics section of your online submission will not be published alongside your manuscript.

Answer: The ethics statement is now located at “Participants and study design” subsection (line 113).

Reviewers' comments:

Reviewer's Responses to Questions

Comments to the Author

1. Is the manuscript technically sound, and do the data support the conclusions?

Reviewer #1: Partly

Reviewer #2: No

2. Has the statistical analysis been performed appropriately and rigorously?

Reviewer #1: No

Reviewer #2: No

3. Have the authors made all data underlying the findings in their manuscript fully available?

Reviewer #1: No

Reviewer #2: No

 Answer: The data are now provided as supporting information.

4. Is the manuscript presented in an intelligible fashion and written in standard English?

Reviewer #1: Yes

Reviewer #2: No

Answer: The first manuscript was revised by a professional English-language editing service. The new version of the manuscript was also fully revised by another professional English-language editing service.

5. Review Comments to the Author

Reviewer #1: PONE-D-20-04410

This is an interesting application of modern psychometric methods. The comparison of IRT models and latent class models appear well suited for the theoretical problem posed. I have some suggestions for the data analysis and presentation of results.

Answer: we thank you for all the provided suggestions. They were insightful and contributed for the work to improve significantly.

1. You should present some basic descriptive information, e.g. the frequency distribution for each item (e.g. Perfect, Imperfect, Wrong), and the frequency distribution of a simple sum of the items (or two sums according to your two approaches). This allows the reader to get some sense of your data.

Answer: we agree with the reviewer and we are thankful for such the suggestion. More information can be found at page 11 Table 1, which provides the percentage of correct answers for Perfect and Imperfect approach for each item, and at page 12 Table 2, which provides the simple correct answers sum for each item for Perfect and Imperfect approach.

2. You estimate a 2-parameter and a 3-parameter IRT model. However, it is unclear which of these models you deem is having the best fit. Also, it was unclear to me, which of these models you compare with the latent class model and for which IRT model you present the score distribution in figure 5. Please clarify.

Answer: We followed your recommendation and we are now using AIC/BIC + likelihood test for difference. Regarding the second raised issue, because there are two criteria for rating, the model comparisons between IRT and latent class analysis are always conducted within the same rate criterion. In the revised manuscript, at page 17, it might be read as follows:

“In terms of a model fit, Table 5 depicts the model fit for IRT models (two and three items parameters) for perfect and imperfect rating. Considering the perfect approach, the lowest BIC was in favor of an IRT with two parameters, whereas for the imperfect approach the lowest BIC was in favor of an IRT with three parameters”

3. Fit is evaluated through a global chi-square test. However, chi-square tests with app. 1000 DF are not optimal. You report χ2(991) = 6635.882, p-value =0.999 (line 239). This must be a typo, if the chisq value is correct, p=0. I suggest you do two things.

3a. For global tests of fit and comparisons of the 2-P and 3-P models, use AIC and BIC as in your comparison with latent class models.

Answer: We agree and reinforce that the comparison was conducted based on AIC and BIC, although the other fit is still reported. 

3b. Evaluate item level fit, e.g. by the item fit tests suggested by Orlando and Thissen (Applied Psychological Measurement 2000). Such tests are available in the IRTPRO software and the free R package mirt (https://cran.r-project.org/web/packages/mirt/mirt.pdf). Such item based fit test may identify some stimuli that are not well modeled by IRT.

Answer: We deeply appreciated the idea about showing the item level fit. We calculated them as described by Orlando and Thissen (2000) using R MIRT package. We added table 4 and a set of figures containing empirical plots for all the items under perfect pitch approach working as illustration. Given these new add-ons, the following information were inserted subheading for dimensional solution (pages 15-16):

“Under IRT with two parameters, imperfect rating indicated that all the items had a good fit (all the S-X² p-values >0.05), whereas, under the perfect approach, two out of the ten items were statistically significant (i.e., items c and d). Under IRT with three parameters, it can be observed that the majority of the items displayed a reduction in their p-values when compared to two parameters, for both approaches. 

One reason why the imperfect approach under two parameters, in terms of item level fit, is better is likely due to the increase in the probabilities of answering the items correctly (i.e., proportion and counts are higher since it is a less strict criterion). In the case of items c and d for a perfect rating, as shown in the empirical plot (Fig 2 and 3), these two items revealed that the closer to the negative amount of theta (AIPWR), the more deviation between what is expected by the model and what is actually being estimated (i.e., confidence interval [in red lines] are far and above from the item characteristic curve [in blue]). Therefore, the observed probability of correctly answering the item at a low spectrum of AIPWR is higher than would be expected.”

4. Your IRT parameter estimates have very large standard errors, in particular for the discrimination parameter in the 3-P model. Difficulties of estimating the discrimination parameter in 3-P models is a known problem, but I am still concerned. You may want to use a prior for the discrimination parameter in addition to a prior for the guessing parameter.

Answer: We understand the concern regarding the SE for the 3-P model. To avoid an extra model (i.e., 3-P model with prior in guessing and discrimination), we added in the Results section the difficulties of estimating the discrimination exactly as you pointed out. Moreover, we gave extra reference about the parsimony rule involving estimating an extra-parameter. 

Now, at page 15, it might be read as follows:

“It is important to note that the standard errors (SE) are larger than under IRT with three parameters, regardless of the adopted rating criterion as commonly described in the literature [40, 41].”

5. I am also a bit worried about the magnitude for the guessing parameter for some items. For pure guessing, you would expect a guessing parameter around 1/12 = 0.08. Is the any theoretical that some items would have a lower asymptote of 0.3?

Answer: The reviewer is correct about lower asymptote of 0.3. This is happening in the imperfect approach. We added an extra information about that point and we made a correction regarding our prior guessing for the imperfect approach, updating it to 3/12 due to the rating criteria where exact pitch responses and semitone deviations were considered correct (e.g.: aural stimulus: C; correct response: C, B or C#/Db). Based on that, we recalculated the prior guessing and the model’s parameters for imperfect rating. We thank you very much for the very attentive reading of our manuscript. Now, at line 207, it might be read as follows:

“Because the keyboard used to report the answers had 12 possibilities, the a priori value for the guessing parameter was 1/12 for the perfect rating; for imperfect rating criteria, it was 3/12 (see below the description for rating)”

6. In choosing between latent class model, you argue that the improved fit of the models with more than 2 classes should be ignored, due to the complexity of these models. However, in comparisons between IRT and latent class models for the perfect approach, you suddenly argue that a difference in fit of the same magnitude is important and should be interpreted. You cannot have it both ways. If you compare e.g. a 3 class model with the IRT model, you get the following results:

AIC D AIC 3CL BIC D BIC 3CL SSABIC D SSABIC 3CL

Perfect 6988.887 6952.444 7082.150 7101.664 7018.640 7000.048

This comparison show no particular superiority of the IRT model in terms of fit. The same reasoning could be applied for a 4-CL or a 5-CL model. You may want to keep a 2-CL model in the comparisons for conceptual reasons, but you should include a latent class model with better fit (e.g. 3 classes or 4 classes). Based on the results I have seen, I would conclude that for the “perfect” approach, a latent class model with 3, 4, or 5 classes has equally good fit as an IRT model. For the “imperfect” approach, the IRT model seems clearly better.

Answer: This was a crucial point. We revised all the Mplus output and we found some inconsistencies in our previous report for AICs and BICs. We are deeply sorry and grateful at that same time, as it made us revise all the numbers and, consequently, our results. Moreover, we added a new suggested modeling approach (hybrid solution). If the reviewer requires Mplus outputs, we would be happy to share with her/him. Now, at page 25 (line 406), our new results are describe as follows:

“Comparing the continuous, categorical, and hybrid solutions we concluded based on model fit information values that the continuous solution was the best solution for perfect approach, with lower BIC than the categorical and hybrid solutions. This indicates that the ability to recognize isolated pitches in different timbers and registers without reference is better modeled as a continuous ability when perfect rating approaches is considered in comparison with a categorical and hybrid model. Alternatively, the categorical solution demonstrated the best solution for the imperfect approach, with lower BIC than the continuous and hybrid solutions. This indicates that adopting flexibility in isolated pitch recognition without reference (with semitone deviations considered as correct) is better modeled as latent groups”

7. You present the score distribution for one IRT model, presumably for the “perfect” approach (the text in lines 272-273 seems to have the numbering of figures wrong). The score distribution is clearly skewed. A large group of people seem to have the same level of ability to identify the pitch of a tone (i.e. no ability). This may pose a problem for standard IRT model estimation, since a normal distribution is assumed for the latent trait. For this reason, a better model for your data may be a latent mixture distribution model with 2 latent classes. Class one consist of persons who are not able to identify the pitch regardless of the stimulus. Class two consists of persons with at least some ability to identify pitch. Within class two responses might follow a 2-P IRT model. Such a model can be estimated in Mplus. It is fairly complex, but you may want to at least discuss it.

Answer: We agree with the reviewer and we added the analysis of a hybrid model. The idea was really nice! 

8. With regards to the psychometric lingo, I would suggest that :

8a. “Psychometric” is better than “Psychometrical”

8b. “Latent variable models” is better than “structural equation model”. SEM refers to a particular type of latent variable models, a type you do not use in your analysis.

8c. “Continuous” is better than “Dimensional”

Answer: We accepted all the corrections and suggestions.

Specific suggestions:

Line 24. I suggest “Through Latent Variable Models (LVM) we can evaluate consistency validity…”

Line 27. I suggest “… two LVM approaches: continuous latent variables (LV)…”

Line 37, I suggest “The phenomenon of absolute pitch (AP)…”

Line 65. It might be helpful to define relative pitch.

Line 115. Is there a reference regarding the use of a combination of AP and RP to identify pitches?

Line 159. I suggest “…continuous and categorical.”

Line 160. I suggest writing “The former approach, Item Response Theory (IRT), …”

Lines 166-170. I suggest writing: “a) An IRT model with two parameters for each stimuli: the discrimination parameter (also called parameter a), which describe the ability of this stimuli to distinguish between persons with low and high pitch identification ability and the item location parameter (also called parameter b), representing the level of pitch identification ability where you have 50% chance of correctly identifying the pitch of this stimulus.”

Lines 172+173. I suggest writing: “… is the probability of a person with very low pitch identification ability still correctly get a correct answer for a given stimulus.

Answer: All the specific points were accepted and a new English-language editing service was contacted to double check spelling, coherence, and fluency.

Reviewer #2: The ultimate aim of this research project is not explicitly stated, and it remains a little unclear. The authors have created a new test of assessing absolute pitch, and this was investigated using an item-response theory (IRT) approach and a latent class analysis (LCA) approach. These models were then compared to see which offered the best fit.

Although the foundation and rationale for the study seems reasonable, in that the authors wish to create a measure to determine whether individuals have the ability to identify absolute pitch, the applied methodologies are confusing and it should be better explained as to why they are appropriate.

Answer: We thank the reviewer for the positive feedback and comments.

• Latent Class Analysis categorises people into groups, under the assumption that the same thing is being measured for all people, by a standardised count, process, or measurement device/scale.

• IRT is used to determine whether a set of items are delivering a valid total score of an unobservable latent trait.

Thus it should be explained in more detail why the fit of these models should be compared as they look at different things. It should be emphasized that the categorisation of people relies on the measurement process being valid and stable, so in some sense LCA should not be considered until the measure has been validated.

Answer: We improved information regarding CFA correcting for model fit and adding item fit indices as below required. For us, it was not clear what the reviewer#2 meant when s/he said about “measurement process being valid and stable, so in some sense LCA should not be considered until the measure has been validated”. Would it mean to conduct a CFA before LCA? CFA/IRT presumes homogeneity in parameter values across cases. LCA implies differences in parameter values, with each case participating in the different populations to a different degree. The hybrid modeling now added, CFA *with* LCA, accounts for heterogeneity by representing the data as a mixture of populations. Granting that LCA is primarily driven by differences in means rather than by differences in variances, we still would not trust parameter estimates (or fit assessment) from a CFA without LCA if we believed that a mixture model was best for the data. Here, we saw exactly that depending on the approach, the best way to model the data will change. 

For the IRT scale assessment approach, there are also many aspects that have been neglected. There is no indication of item fit. No investigation of response dependency. No assessment of reliability or targeting.

Answer: This point was also raised by reviewer#1 and now, at page 9, it might be read as follows: 

“To evaluate the model fit indices for IRT models, a Pearson chi-square test for categorical outcomes was used, with p-values higher than 0.05 indicators of a good fit. Item level fit was evaluated via Pearson’s X2 (S-X2) implemented in R package MIRT, as per Orlando and Thissen [34]”

Was a single parameter model considered? - A single-parameter (Rasch) model would be appropriate for scale development and validation purposes, and for determining whether a total score from a set of items is a sufficient statistic to assess the level of a latent trait.

Answer: We did not consider a single-parameter model leaving only 2 and 3 parameters. We did not find references discussing the superiority of 2/3 parameters modeling in comparison to single-parameter modeling for scale development and validation purposes. Based on the Deborah Bandalos 2018 book, on the chapter on Validity, it is stated that evidence based on internal consistency (i.e., commonly called as construct validity) are achieved by techniques involving latent variable approach not directly specifying the number of parameters to be estimated. 

Additionally, for the ‘imperfect approach’, it may be worth the authors considering a partial-credit model, where an exact pitch classification is awarded a score of 2, a semitone deviation is awarded 1, and all other pitches are scored 0.

Answer: We understand the point raised by the reviewer and we understand the underlying idea of given this graded answer to the tasks. However, due to two main issues, we decide not to conduct it this way. Firstly, in the absolute pitch area, such scoring system (2, 1, 0) is not commonly adopted. Second, after the new added model (hybrid) plus other adds-on, we believe that the manuscript increased in its complexity. Adding a new modeling, which is only sensitive/meaningful for imperfect approach, will increase even more the complexity and length of our manuscript. This idea will be for sure further explored in future works. 

There are also some further issues within the manuscript that would need attention:

The model fit statistics are dubious, and there is no real interpretation of the fit statistics that are presented. Certainly a test with 1000 degrees of freedom will have no statistical power.

Answer: We excluded the chi-square test for 1000 degrees of freedom and we improved the interpretability of the figure and new/old tables.

In the manuscript, it is stated that items are centred around the 0 location, but there are no item locations reported below 0 – where are they centred?

Answer: We are sorry for our imprecision. Now, at page 9, it might be read as follows: 

“Under Maximum Likelihood estimator and using logit parameterization (theta), the constant 1.7 in the logit gives only an approximate closeness to the normal. The translation to IRT parameter values uses factor mean and factor variance to bring them to the N~(0,1) metric used in IRT”

The pitch test is based across different musical instruments – have these instruments been calibrated? Has the pitch been externally verified in some way?

Answer: The samples used were professionally recorded in a studio and are extremely accurate in terms of tuning, based on the A=440Hz standard. Since they are commercially used for a variety of soundtracks in cinema and television, their precision and quality is known among professional musicians. Nevertheless, the pitch of every sample used was checked before the battery test was assembled. The voice samples were recorded and edited by a professional studio technician. Two professional vocalists were assigned for the task. The pitches of the recorded voice samples were also checked (A=440Hz standard) and any deviation was corrected.

Additionally, the manuscript is currently in need of a language edit and the Figures are incorrectly labelled.

 Answer: We agree with the reviewer that a language review in the whole manuscript was needed. We sent it to a new professional Editing-language editing service for a fully review. We deeply thank you for all the recommendations.

6. PLOS authors have the option to publish the peer review history of their article (what does this mean?). If published, this will include your full peer review and any attached files.

Do you want your identity to be public for this peer review? For information about this choice, including consent withdrawal, please see our Privacy Policy.

Reviewer #1: Yes: Jakob Bue Bjorner

Reviewer #2: No

---

## [Decision Letter · Decision Letter 1]

8 Sep 2020

PONE-D-20-04410R1

A new Approach to Measuring Absolute Pitch on a Psychometric Theory of Isolated Pitch Perception: Is it Disentangling two Groups or Capturing a Continuous Ability?

PLOS ONE

Dear Dr. Germano,

Thank you for submitting your manuscript to PLOS ONE. After careful consideration, we feel that it has merit but does not fully meet PLOS ONE’s publication criteria as it currently stands. Therefore, we invite you to submit a revised version of the manuscript that addresses the points raised during the review process.

Thank you for your careful revision. I agree with the reviwers that the manuscript has improved. The comments from the reviewers on the revised version illustretes that more work is needed. I agree with comment that the manuscript is currently too long and too difficult to read. I am also concerned about the results you obtain from the 3PL model. The very large standard errors, and the large guessing parameters makes me worry that these results cannot be trusted. One way to make the manuscript better would be to put less emphasis on these results.

Please carefully consider all the points raised in the attached reviews and submit your revised manuscript by Oct 23 2020 11:59PM. If you will need more time than this to complete your revisions, please reply to this message or contact the journal office at plosone@plos.org. Please include the following items when submitting your revised manuscript:

We look forward to receiving your revised manuscript.

Kind regards,

Karl Bang Christensen, Ph.D.

Academic Editor

PLOS ONE

Reviewers' comments:

Reviewer's Responses to Questions

**Comments to the Author**

1. If the authors have adequately addressed your comments raised in a previous round of review and you feel that this manuscript is now acceptable for publication, you may indicate that here to bypass the “Comments to the Author” section, enter your conflict of interest statement in the “Confidential to Editor” section, and submit your "Accept" recommendation.

Reviewer #1: (No Response)

Reviewer #2: All comments have been addressed

2. Is the manuscript technically sound, and do the data support the conclusions?

Reviewer #1: No

Reviewer #2: Yes

3. Has the statistical analysis been performed appropriately and rigorously? 

Reviewer #1: Yes

Reviewer #2: I Don't Know

4. Have the authors made all data underlying the findings in their manuscript fully available?

Reviewer #1: No

Reviewer #2: Yes

5. Is the manuscript presented in an intelligible fashion and written in standard English?

Reviewer #1: No

Reviewer #2: Yes

6. Review Comments to the Author

Reviewer #1: Thanks for the responses to my previous comments. I find that your revisions have clarified the data analyses, but there is still some way to go.

You have two scoring options for assessment of isolated pitch perception: a. Perfect: only the correct note is classified as a correct response, b. Imperfect: including the half-note below and above. You compare 1. IRT analyses using either a 2-PL or a 3-PL model, 2. Latent class analyses, 3. Mixture model analyses. You conclude that perfect scoring is best fitted using an IRT model, while imperfect scoring is best fitted with a two-class latent class model.

1. I understand that your choice of model is guided by the global indices such as BIC and AIC. However, from a conceptual point of view, this conclusion does not make sense to me. If you conclude that isolated pitch perception with perfect scoring represent a continuous latent trait, how could the use of a less stringent scoring criterion suddenly change this ability to two latent classes? It seems to me that this interpretation of results is theoretically incoherent. I think you need to discuss this and make an overall interpretation whether isolated pitch perception is best considered a continuous trait or two latent classes.

2. Related to the discussion above, it seem to me that the latent class solution for imperfect scoring has some unfortunate interpretation. Even the best group has only a 67% chance of getting item i right ad 72% change of getting item j right within +/- a half tone. This does not concur with the common understanding of absolute pitch. Maybe the absolute pitch group is only a subgroup within the current best latent class. In that case, you may need more than two latent classes. Please discuss whether your current model is theoretically plausible.

3. For perfect scoring, you find that an IRT model represent the best model for the data. However, for perfect scoring, evaluation of item fit finds significant misfit for item c and item d. This suggests problems for these two items as indicators of the latent trait. This is not discussed, but should be. This plots and fit tests do not suggest that a 3-PL model provides a better fit to the data than a 2-PL model.

4. I continue having concerns about your 3-PL model. The discrimination parameter is very high for item b and so it the guessing parameter. Also, the standard errors of the discrimination parameter are high. The BIC suggest that the 2-PL model might be the best. You should conclude which model you regard as the final model.

5. You aim to develop a test for isolated pitch recognition. I assume that part of this development is to decide whether your test is best scored using the perfect or the imperfect approach. I think you should provide your recommendation and the reasoning behind it.

6. The paper is a long and complex read because some many combinations of options are examined. You would increase readability by focusing the main paper on what you consider the best solution and present other options as supplemental analyses. For example, you could focus on perfect scoring, use the 2-PL model as your IRT model to be compared with a latent class and a mixture analysis. Other options could be alluded to briefly and results for these other models could be presented in a web appendix. I think such an approach would make your paper much more readable.

7. You write that the differences in item difficult and discrimination poses difficulties for a simple sum score approach. However, items may be summed without problems even if they have widely different item difficulty. For example, in the Rasch model (where all items have the same discrimination, but may differ in item difficulty) the sum score is a sufficient statistics for the latent trait. So the real issue is whether the items very so much in item difficulty that a simple sum is inappropriate. For the 2-PL model and perfect scoring, I do not think this is the case. Item discrimination varies between 1.2 and 1.9, not a dramatic variation. It is possible that a Rasch (i.e. 1-PL model) may fit these items. Please revise this discussion.

8. While the analyses seems to be well done, the interpretation and discussion of results could use input from English language researchers with psychometric / IRT expertise. Some description of the models is not well structured (e.g. the discussion of IRT models on lines 205-219). Also, while most of the paper is well written, some parts of the psychometric discussion still deviates for the normal language of the field, e.g. in the abstract [my suggestions in square brackets]:

“We decided to adopt a psychometric perspective, approaching AP as a latent trait. Via Latent Variable Model (LVM) we can provide [evaluate] consistency and validity for a measurement [measure] to test for AP ability. A total of 783 undergraduate music students took part in the test. The battery test [test battery] consisted of 10 isolated pitches.”

Reviewer #2: I would like to thank the authors for responding to the reviewers’ requests and making appropriate amendments to their paper. They have clearly invested time and effort into this, and I believe that the manuscript is now presented better and is much clearer in terms of the purpose of the paper and the process that has been carried out.

There are a few additional amendment suggestions that I have, and these are provided below:

I would suggest that the title is amended to state ‘specific groups’ or ‘ability groups’ rather than ‘two groups’. As you are using LCA, it is not known a priori how many latent classes will be identified.

The Intro reads well, provides good background and makes sense. However, this is an exploratory study, to see how the items in the AP test work among the group tested. Do the items work together to form a measure of an underlying latent continuum (IRT)? Or do they work better as a set of indicator items that can classify people into groups (LCA)? I would suggest that this may be clarified for readers if the authors were to provide a statement in both the abstract and the introduction to state that this is an exploratory data modelling study, to determine which type of model best fits the data for the tested sample.

For the IRT analysis, there is still no test of local dependency among the items. This is perhaps unnecessary for the purpose of the current study, but perhaps it could be identified as a potential limitation, or the authors could suggest that it could be assessed in future work if a latent trait IRT approach is pursued further.

The authors state ‘Under Maximum Likelihood estimator and using logit parameterization (theta), the constant 1.7 in the logit gives only an approximate closeness to the normal. The translation to IRT parameter values uses factor mean and factor variance to bring them to the N~(0,1) metric used in IRT.’ To clarify this for the reader, I would suggest that the authors might also add a sentence to state that this means that the IRT analysis is centred on the person sample being at 0 logits, and that the item difficulty parameters are provided relative to this.

A few additional very minor corrections are as follows:

Line 189 states that MPlus is used. R also needs to be added here.

Line 287. I believe this should say difficulty rather than discrimination.

Line 409. Should be timbres rather than timbers.

A list of abbreviations would also be useful, so that the reader can refer back to them without scrolling through all of the manuscript to find the relevant abbreviation.

7. PLOS authors have the option to publish the peer review history of their article (what does this mean?). If published, this will include your full peer review and any attached files.

Reviewer #1: **Yes: **Jakob Bue Bjorner

Reviewer #2: No

---

## [Author Response · Author response to Decision Letter 1]

21 Dec 2020

PONE-D-20-04410R1

A new Approach to Measuring Absolute Pitch on a Psychometric Theory of Isolated Pitch Perception: Is it Disentangling two Groups or Capturing a Continuous Ability?

PLOS ONE

Dear Dr. Germano,

Thank you for submitting your manuscript to PLOS ONE. After careful consideration, we feel that it has merit but does not fully meet PLOS ONE’s publication criteria as it currently stands. Therefore, we invite you to submit a revised version of the manuscript that addresses the points raised during the review process.

Thank you for your careful revision. I agree with the reviwers that the manuscript has improved. The comments from the reviewers on the revised version illustretes that more work is needed. I agree with comment that the manuscript is currently too long and too difficult to read. I am also concerned about the results you obtain from the 3PL model. The very large standard errors, and the large guessing parameters makes me worry that these results cannot be trusted. One way to make the manuscript better would be to put less emphasis on these results.

Please carefully consider all the points raised in the attached reviews and submit your revised manuscript by Oct 23 2020 11:59PM. If you will need more time than this to complete your revisions, please reply to this message or contact the journal office at plosone@plos.org. Please include the following items when submitting your revised manuscript:

We look forward to receiving your revised manuscript.

Kind regards,

Karl Bang Christensen, Ph.D.

Academic Editor

PLOS ONE

Reviewers' comments:

Reviewer's Responses to Questions

Comments to the Author

1. If the authors have adequately addressed your comments raised in a previous round of review and you feel that this manuscript is now acceptable for publication, you may indicate that here to bypass the “Comments to the Author” section, enter your conflict of interest statement in the “Confidential to Editor” section, and submit your "Accept" recommendation.

Reviewer #1: (No Response)

Reviewer #2: All comments have been addressed

2. Is the manuscript technically sound, and do the data support the conclusions?

Reviewer #1: No

Reviewer #2: Yes

3. Has the statistical analysis been performed appropriately and rigorously?

Reviewer #1: Yes

Reviewer #2: I Don't Know

4. Have the authors made all data underlying the findings in their manuscript fully available?

Reviewer #1: No

Reviewer #2: Yes

5. Is the manuscript presented in an intelligible fashion and written in standard English?

Reviewer #1: No

Reviewer #2: Yes

Answer: The first and second manuscript was revised by a professional English-language editing service. The new version of the manuscript was also fully revised by another professional English-language editing service.

6. Review Comments to the Author

Reviewer #1: Thanks for the responses to my previous comments. I find that your revisions have clarified the data analyses, but there is still some way to go.

You have two scoring options for assessment of isolated pitch perception: a. Perfect: only the correct note is classified as a correct response, b. Imperfect: including the half-note below and above. You compare 1. IRT analyses using either a 2-PL or a 3-PL model, 2. Latent class analyses, 3. Mixture model analyses. You conclude that perfect scoring is best fitted using an IRT model, while imperfect scoring is best fitted with a two-class latent class model.

1. I understand that your choice of model is guided by the global indices such as BIC and AIC. However, from a conceptual point of view, this conclusion does not make sense to me. If you conclude that isolated pitch perception with perfect scoring represent a continuous latent trait, how could the use of a less stringent scoring criterion suddenly change this ability to two latent classes? It seems to me that this interpretation of results is theoretically incoherent. I think you need to discuss this and make an overall interpretation whether isolated pitch perception is best considered a continuous trait or two latent classes.

Response: Thank you for your observation. AP literature shows different methods to measure AP ability, which led to the two rating criteria (perfect and imperfect approaches) adopted in our test. To clarify, we conducted a brief review of the two methods (see table below), showing that both methods are largely used universally. Our aim was not to directly compare these two approaches. Rather, we aimed to evaluate how they might influence the decision of the best model for isolated pitch recognition tasks (line 111), once they are both used arbitrarily in the literature. One of our hypotheses was that different measurement methods lead to different results that reflect different theoretical perspectives. This is a methodological issue that is not adequately addressed by AP literature. Our results corroborate this hypothesis, as the two rating approaches showed distinct underlying models (a continuous trait for the perfect approach and two latent classes for the imperfect approach). Also, our aim was to contribute to a better theoretical understanding of AP ability, showing that different rating criteria greatly influences test results and the underlying model. We added the imperfect result in the abstract to highlight this duality regarding AP measurement (line 37). In the conclusion, we added our contribution to the AP literature (line 511). In the discussion, we added an observation regarding the change from a continuous model to a two latent classes model resulting from the adoption of a less stringent scoring criterion, as mentioned (line 466).

2. Related to the discussion above, it seem to me that the latent class solution for imperfect scoring has some unfortunate interpretation. Even the best group has only a 67% chance of getting item i right ad 72% change of getting item j right within +/- a half tone. This does not concur with the common understanding of absolute pitch. Maybe the absolute pitch group is only a subgroup within the current best latent class. In that case, you may need more than two latent classes. Please discuss whether your current model is theoretically plausible.

Response: 

We thank the reviewer for raising this issue. 

AP possessors do not have exceptional pitch acuity. Absolute pitch is neither ‘absolute’ nor ‘perfect’ in the ordinary uses of those words (Levitin e Rogers, 2005). AP literature shows that it is quite common for AP possessors to make semitone errors in pitch judgments (Rogers and Levitin, 2005; Miyazaki, 1988; Baggaley, 1974; Brady, 1970). This results in semitone errors being considered as a partial or totally correct answer in some tests (Baharloo et al., 1998; Athos et al., 2007; Schulze et al., 2009). In our study, under the imperfect approach, the better group had a 67% chance of getting item i right and a 72% chance of getting item j right. These results are consistent with previous research using the imperfect approach, considering semitone errors as ½ correct (73% correct), not allowing for semitone errors and 80% correct allowing for semitone errors, [Li (2020]). 

Therefore, contrastingly to what Reviewer#1 mentioned, and as described by Levitin and Rogers (2005), page 28, “... ‘absolute’ refers to judgments established independently, rather than by comparison. The terms ‘absolute’ and ‘perfect’ both imply in the lay mind a level of precision not typically present in AP possessors, who frequently make octave errors (confusing tones that are half or double the frequency), and semitone errors (confusing tones that are 6% apart) [Levitin, D.J. (1999) Absolute pitch: Self-reference and human memory. International Journal of Computing Anticipatory Systems 4, 255–266–Miyazaki, K. (1988) Musical pitch identification by absolute pitch possessors. Percept. Psychophys. 44, 501–512]. Like most human traits, AP is not an all-or-none ability, but rather, exists along a continuum [Levitin, D.J. (2004) L’oreille absolue. L’Annee´ Psychologique 104, 103–120; Levitin, D.J. (1999) Absolute pitch: Self-reference and human memory. International Journal of Computing Anticipatory Systems 4, 255–266; Deutsch, D. (2002) The puzzle of absolute pitch. Curr. Dir. Psychol. Sci. 11, 200–204; Vitouch, O. (2003) Absolutist models of absolute pitch are absolutely misleading. Music Perception 21, 111–117]. Self-identified AP possessors score well above chance (which would be 1 out of 12, or 8.3%) on AP tests, typically scoring between 50 and 100% correct [Miyazaki, K. (1988) Musical pitch identification by absolute pitch possessors. Percept. Psychophys. 44, 501–512], and even musicians not claiming AP score up to 40% [Lockhead, G.R. and Byrd, R. (1981) Practically perfect pitch. J. Acoust. Soc. Am. 70, 387–389]. Still, even those who score better than 90% show similar discrimination thresholds to, and are typically no better than, other musicians at noticing when one tone is out of tune with respect to another [Burns, E.M. and Campbell, S.L. (1994) Frequency and frequency-ratio resolution by possessors of absolute and relative pitch: Examples of categorical perception? J. Acoust. Soc. Am. 96, 2704–2719; Levitin, D.J. (1999) Absolute pitch: Self-reference and human memory. International Journal of Computing Anticipatory Systems 4, 255–266]. Clearly, there is nothing ‘perfect’ about AP; rather AP is the ability to place or produce tones within nominal categories.

This discussion was included in line 96. We also conducted a three latent class analysis, but the two classes solution was best, as presented in line 379. If there were volunteers who were near infallible in isolated pitch recognition tasks in our experiment, they could not be identified as a separate group among the participants. This discussion was added in line 466.

3. For perfect scoring, you find that an IRT model represent the best model for the data. However, for perfect scoring, evaluation of item fit finds significant misfit for item c and item d. This suggests problems for these two items as indicators of the latent trait. This is not discussed, but should be. This plots and fit tests do not suggest that a 3-PL model provides a better fit to the data than a 2-PL model.

Response: We thank the Reviewer for this suggestion. We agree with the Reviewer, especially considering the parsimony principle. We modified our text, accordingly stating that the 2-PL is the best solution for both ratings under the IRT approach. 

4. I continue having concerns about your 3-PL model. The discrimination parameter is very high for item b and so it the guessing parameter. Also, the standard errors of the discrimination parameter are high. The BIC suggest that the 2-PL model might be the best. You should conclude which model you regard as the final model.

Response: Thank you for your considerations. The 2-PL model was concluded as the best model. The text modification was made in line 379-384 and in line 461.

5. You aim to develop a test for isolated pitch recognition. I assume that part of this development is to decide whether your test is best scored using the perfect or the imperfect approach. I think you should provide your recommendation and the reasoning behind it.

Response: As previously discussed, our aim was not to compare these two score approaches (because the models are not nested). We sought instead to evaluate how they might influence the decision of the best model for isolated pitch recognition tasks (line 112). As score approaches reflect different theoretical perspectives, we believe our results contribute to a better theoretical understanding of AP ability, showing that different rating criteria greatly influences test results and the nature of the underlying latent variable. Also, we cannot formally compare imperfect and perfect approaches in regarding superiority because they are not nested models (line 453). 

The test comprised auditory stimuli and was collectively applied. Volunteers were given a keyboard drawn on paper on which they marked the answer that they thought was correct. This procedure allowed us to standardize the evaluation procedure. The correction was conducted in two ways (perfect and imperfect approaches), according to the literature. 

Given below are examples of studies that use these two approaches:

Auditory Stroop and Absolute Pitch: An fMRI Study - Katrin Schulze, Karsten Mueller, and Stefan Koelsch Human Brain Mapping

Imperfect (answers within one semitone were regarded as a correct answer)

Intracortical Myelination in Musicians With Absolute Pitch: Quantitative Morphometry Using 7-T MRI -Seung-Goo Kim and Thomas R. Kn€osche - Human Brain Mapping - Imperfect (An error with a semitone was considered as a correct response)

Increased Volume and Function of Right Auditory Cortex as a Marker for Absolute Pitch- Martina Wengenroth, Maria Blatow, Armin Heinecke, Julia Reinhardt, Christoph Stippich, Elke Hofmann and Peter Schneider- Cerebral Cortex - Imperfect (for semitone errors 0.5 point was Accredited)

Gray and White Matter Anatomy of Absolute Pitch Possessors - Anders Dohn, Eduardo A. Garza-Villarrea, M. Mallar Chakravarty,

Mads Hansen, Jason P. Lerch, and Peter Vuust- Cerebral Cortex -Imperfect (Participants were given ¾ point for each error of a semitone)

Perceiving pitch absolutely: Comparing absolute and relative pitch possessors in a pitch memory task- Katrin Schulze, Nadine Gaab and Gottfried Schlaug BMC Neuroscience - Imperfect (answers within one semitone of the sented pitch as a correct answer)

The Neurocognitive Components of Pitch Processing: Insights from Absolute Pitch Sarah J. Wilson, Dean Lusher, Catherine Y. Wan,

Paul Dudgeon and David C. Reutens Cerebral Cortex - Perfect (semitone errors were coded as incorrect for all participants)

Absolute Pitch and the P300 Component of the Event-Related Potential: An Exploration of Variables That May Account for Individual Differences - Laura Renninger, Roni Granot, Emanuel Donchin - Music Perception - Imperfect (credit was given to subject was came within 1 semitone of the correct pitch)

Absolute Pitch—Functional Evidence of Speech-Relevant Auditory Acuity Mathias S. Oechslin, Martin Meyer and Lutz Jäncke - Cerebral Cortex - Perfect (the semitone errors were taken as incorrect to increase the discriminatory power)

Absolute Pitch and Planum Temporale- Julian Paul Keenan, Ven Thangaraj, Andrea Halpern, Gottfried Schlaug NeuroImage - Imperfect (we regarded a response within 1/2 tone difference of the presented tone as a correct response)

Absolute Pitch: An Approach for Identification of Genetic and Nongenetic Components- Siamak Baharloo, Paul A. Johnston, Susan K. Service, Jane Gitschier, and Nelson B. Freimer Am. J. Hum. Genet. Imperfect (we decided to score a full point for semitone errors made by individuals >45 years of age)

A Distribution of Absolute Pitch Ability as Revealed by Computerized Testing- Patrick Bermudez And Robert J. Zatorre Music Perception - Perfect (In the percent correct score, only exactly correct responses are counted (0 semitone deviation)

Absolute pitch is associated with a large auditory digit span: A clue to its genesis (L)- Diana Deutsch and Kevin Dooley J. Acoust. Soc. Am. Perfect (not allowing for semitone errors).

Absolute pitch among American and Chinese conservatory students: Prevalence differences, and evidence for a speech-related critical period - Diana Deutsch, Trevor Henthorn, Elizabeth Marvin, HongShuai Xi J. Acoust. Soc. Am. - Perfect and Imperfect (no semitone errors allowed and semitone errors allowed).

Dichotomy and perceptual distortions in absolute pitch ability - Alexandra Athos, Barbara Levinson, Amy Kistler, Jason Zemansky, Alan Bostrom, Nelson Freimer, and Jane Gitschier PNAS - Imperfect (partial [3/4 point] credit for an answer deviating by one semitone

Effects of Musical Training and Absolute Pitch on a Pitch Memory Task: an Event-related Potential Study- Edwin C Hantz, Kelley G. Kreilick, Amy L. Braveman, Kenneth P. Swartz Psychomusicology - Perfect (no points were given for any other answers; that is, no points were awarded for near hits half-step errors)

The effects of timbre on absolute pitch judgment Xiaonuo Li Psychology of Music (new research)- Perfect and Imperfect (no semitone errors allowed and semitone errors allowed).

Absolute Pitch as an Inability: Identification of musical interval in a tonal context- Ken’ichi Miyazaki- Musical Perception - Imperfect (semitone errors are counted as correct)

Perfect Pitch - Joseph Profita and T. George Bidder - American Journal of Medical Genetics - Perfect (Subjects were defined as having perfect pitch if they were able to identify 90% or more of the total number of tones).

Absolute Pitch and the P300 Component of the Event-Related Potential: An Explanatory of Variables that may Account for Individuals Differences - Laura Bischoff Renninger, Roni Granot, Emanuel Donchin- Music Perception - Imperfect (credit was given to subject who came within 1 semitone of the correct pitch)

Absolute Pitch: Effects of Timbre on Note-Naming Ability- Patrícia Vanzella, E. Glenn Schellenberg PLoS ONE - Imperfect (semitones errors were considered as correct)

Effects of musical training and absolute pitch ability on event-related activity in response to sine tones - John W. Wayman, Robert D. Frisina and Joseph P. Walton - Acoustical Society of America - Imperfect (half-step errors were given half credit)

How Stable is Pitch Labeling Accuracy in Absolute Pitch Possessors? - Wilfried Gruhn, Reet Ristmägi, Peter Schneider, Arun D'souza, Kristi Kiilu- Empirical Musicological Review - Imperfect (For semitone errors 0.5 point was accredited)

Absolute pitch memory: Its prevalence among musicians and dependence on the testing context- Yetta Kwailing Wong & Alan C.N. Wong - Psychon Bull Rev - Perfect (discriminating between neighboring semitones [e.g., treating a “C” as a “C#”] are regarded as errors) (Takeuchi & Hulse, 1993; Zatorre, 2003). 

Multiple coding strategies in the retention of musical tones by possessors of absolute pitch - Robert J. Zatorre, Christine Beckett Memory & Cognition - Perfect (0% correct by semitone transposition)

6. The paper is a long and complex read because some many combinations of options are examined. You would increase readability by focusing the main paper on what you consider the best solution and present other options as supplemental analyses. For example, you could focus on perfect scoring, use the 2-PL model as your IRT model to be compared with a latent class and a mixture analysis. Other options could be alluded to briefly and results for these other models could be presented in a web appendix. I think such an approach would make your paper much more readable.

Response: Thank you for your comment. We agree that this paper is a long and complex read. Nevertheless, all these analyses are of paramount importance to researchers in the AP area. We believe that the audience and the impact of this paper will be expanded if these two scoring approaches commonly cited and used in AP literature are included in the main text. Please see the table above for your reference where we mapped the manuscripts, number of citations, the adopted approaches, and other bibliometric features. Applying psychometrics to AP may guide future studies, since our work provides different analytical perspectives for a set of stimuli commonly used to track AP. That is, if our paper addresses these two different approaches (imperfect versus perfect), it will be increasingly cited in the future due to the novel AP research. As mentioned above, the two score approaches reflect different theoretical perspectives which are commonly adopted. We believe that this full comparative and statistical discussion will contribute to a better theoretical understanding of AP ability, showing that different rating criteria greatly influences test results and the underlying model.

7. You write that the differences in item difficult and discrimination poses difficulties for a simple sum score approach. However, items may be summed without problems even if they have widely different item difficulty. For example, in the Rasch model (where all items have the same discrimination, but may differ in item difficulty) the sum score is a sufficient statistics for the latent trait. So the real issue is whether the items very so much in item difficulty that a simple sum is inappropriate. For the 2-PL model and perfect scoring, I do not think this is the case. Item discrimination varies between 1.2 and 1.9, not a dramatic variation. It is possible that a Rasch (i.e. 1-PL model) may fit these items. Please revise this discussion.

Response: We agree with the Reviewer. The process of parceling (summing or averaging items) has shown its robustness based on the law of large numbers and aggregation principles (see Little et al., 2002). The sentence has been removed from the text.

Little, T. D., Cunningham, W. A., Shahar, G., & Widaman, K. F. (2002). To parcel or not to parcel: Exploring the question, weighing the merits. Structural equation modeling, 9(2), 151-173.

8. While the analyses seems to be well done, the interpretation and discussion of results could use input from English language researchers with psychometric / IRT expertise. Some description of the models is not well structured (e.g. the discussion of IRT models on lines 205-219). Also, while most of the paper is well written, some parts of the psychometric discussion still deviates for the normal language of the field, e.g. in the abstract [my suggestions in square brackets]:

“We decided to adopt a psychometric perspective, approaching AP as a latent trait. Via Latent Variable Model (LVM) we can provide [evaluate] consistency and validity for a measurement [measure] to test for AP ability. A total of 783 undergraduate music students took part in the test. The battery test [test battery] consisted of 10 isolated pitches.”

Response: Thank you for your suggestions. We changed the words in the abstract and in the lines indicated. 

Reviewer #2: I would like to thank the authors for responding to the reviewers’ requests and making appropriate amendments to their paper. They have clearly invested time and effort into this, and I believe that the manuscript is now presented better and is much clearer in terms of the purpose of the paper and the process that has been carried out.

There are a few additional amendment suggestions that I have, and these are provided below:

I would suggest that the title is amended to state ‘specific groups’ or ‘ability groups’ rather than ‘two groups’. As you are using LCA, it is not known a priori how many latent classes will be identified.

Response: Thank you for the suggestion. We changed “two groups” to “specific groups”.

The Intro reads well, provides good background and makes sense. However, this is an exploratory study, to see how the items in the AP test work among the group tested. Do the items work together to form a measure of an underlying latent continuum (IRT)? Or do they work better as a set of indicator items that can classify people into groups (LCA)? I would suggest that this may be clarified for readers if the authors were to provide a statement in both the abstract and the introduction to state that this is an exploratory data modelling study, to determine which type of model best fits the data for the tested sample.

Response: Thank you for your comments. AP literature indicates different approaches to rating the stimuli of isolated pitch perception, which led to the two rating criteria (the perfect and imperfect approaches) adopted in our test. Our aim was not to compare these two approaches, but to evaluate how they might influence the decision of the best model for isolated pitch recognition tasks within each approach (line 112). One of our hypotheses was that different approaches lead to different results. It is important to note that both approaches are used in AP research and, to our knowledge, this psychometric issue was not adequately addressed by AP literature. Our results corroborate this hypothesis, as the two rating approaches showed the distinct nature of latent variables (a continuous trait for the perfect approach and two latent classes for the imperfect approach). Also, our aim was to contribute to a better theoretical understanding of AP ability, showing that different rating criteria greatly influence test results and how the latent variable might be measured. We agree that this issue needed further clarification in the manuscript. Therefore, we included more information in lines 96, 466 and 509.

For the IRT analysis, there is still no test of local dependency among the items. This is perhaps unnecessary for the purpose of the current study, but perhaps it could be identified as a potential limitation, or the authors could suggest that it could be assessed in future work if a latent trait IRT approach is pursued further.

Response: We agree with the Reviewer. We added the following: “Future studies may investigate more detailed elements of psychometrics as local dependency for each of the models (IRT and LCA), invariance testing per sex, time of studies, and played instruments” (lines 496 – 498)

The authors state ‘Under Maximum Likelihood estimator and using logit parameterization (theta), the constant 1.7 in the logit gives only an approximate closeness to the normal. The translation to IRT parameter values uses factor mean and factor variance to bring them to the N~(0,1) metric used in IRT.’ To clarify this for the reader, I would suggest that the authors might also add a sentence to state that this means that the IRT analysis is centred on the person sample being at 0 logits, and that the item difficulty parameters are provided relative to this.

Response: We thank the Reviewer for this comment. It now reads in line 216 as: “The factor is assumed to be normally distributed being the mean fixed at zero and factor variance at 1. That is, the IRT analysis is centered on the person sample being at 0 logits, and that the item difficulty parameters are provided relative to this” 

A few additional very minor corrections are as follows:

Line 189 states that MPlus is used. R also needs to be added here.

Response: Thank you for the observation. The R program was included. 

Line 287. I believe this should say difficulty rather than discrimination.

Response: Thank you. The word has been changed.

Line 409. Should be timbres rather than timbers.

Response: You are correct. The revision has been made. 

A list of abbreviations would also be useful, so that the reader can refer back to them without scrolling through all of the manuscript to find the relevant abbreviation.

Response: Thank you for your suggestion. The list of abbreviations was included as Supporting Information. 

7. PLOS authors have the option to publish the peer review history of their article (what does this mean?). If published, this will include your full peer review and any attached files.

Do you want your identity to be public for this peer review? For information about this choice, including consent withdrawal, please see our Privacy Policy.

Reviewer #1: Yes: Jakob Bue Bjorner

Reviewer #2: No

[NOTE: If reviewer comments were submitted as an attachment file, they will be attached to this email and accessible via the submission site. Please log into your account, locate the manuscript record, and check for the action link "View Attachments". If this link does not appear, there are no attachment files.

---

## [Decision Letter · Decision Letter 2]

9 Feb 2021

A new approach to measuring absolute pitch on a psychometric theory of Isolated Pitch Perception: Is it disentangling specific groups or capturing a continuous ability?

PONE-D-20-04410R2

Dear Dr. Germano,

We’re pleased to inform you that your manuscript has been judged scientifically suitable for publication and will be formally accepted for publication once it meets all outstanding technical requirements.

Kind regards,

Karl Bang Christensen, Ph.D.

Academic Editor

PLOS ONE

Additional Comments:

Please edit the manuscript according to these helpful comments from the two reviewers listed below. 

**Comments to the Author**

1. If the authors have adequately addressed your comments raised in a previous round of review and you feel that this manuscript is now acceptable for publication, you may indicate that here to bypass the “Comments to the Author” section, enter your conflict of interest statement in the “Confidential to Editor” section, and submit your "Accept" recommendation.

Reviewer #1: All comments have been addressed

Reviewer #2: All comments have been addressed

2. Is the manuscript technically sound, and do the data support the conclusions?

Reviewer #1: Yes

Reviewer #2: Yes

3. Has the statistical analysis been performed appropriately and rigorously? 

Reviewer #1: Yes

Reviewer #2: Yes

4. Have the authors made all data underlying the findings in their manuscript fully available?

Reviewer #1: Yes

Reviewer #2: Yes

5. Is the manuscript presented in an intelligible fashion and written in standard English?

Reviewer #1: Yes

Reviewer #2: Yes

6. Review Comments to the Author

Reviewer #1: Thanks for the responses to my previous comments. I find that the manuscript is further improved. I only have some suggestions for improvement of language.

1. Line 193. I suggest writing “and the R program” and cite e.g.: R Core Team (2020). R: A language and environment for statistical computing. R Foundation for Statistical Computing, Vienna, Austria. URL https://www.R-project.org/.

2. Line 194-195. I suggest writing: “The former Item Response Theory (IRT) approach …”

3. Line 198: I suggest writing: “Two different IRT models were used”

4. Line 223: I suggest writing: “Pearson’s X2 (S-X2) implemented in the R package mirt, as per Orlando and Thissen [38]”

5. Line 263, I suggest writing: “given that there were three approaches to statistical modeling…”

6. Lines 315-321: I do not agree with your comments on figure 2 and 3. There are indication of both overestimation and underestimation for low score levels. I suggest just stating: “For items c and d – scored with the criterion of perfect rating – misfit is illustrated by comparisons of predicted and observed proportion of correct results (Fig 2 and 3). In particular, higher than expected proportion of correct answers are seen for theta scores a little higher than 1 and for theta scores a little lower than -1.”

7. Line 417, I suggest writing: “Based on model fit information, we conclude that the continuous…”

8. Line 446, I suggest writing: ”Moreover, in a two-parameter IRT model for the perfect scoring approach, all the items showed…”

9. Line 454-455, I suggest dropping the first part of the sentence and just write: “When comparing LCA to IRT …”

10. Line 472-476, You write “If there are participants who are near infallible in isolated pitch recognition tasks, their prevalence will be reduced as the scores increases (i.e., the higher the score, the lower the number of subjects endorsing all the stimuli correctly). However, under the imperfect approach, all the participants that committed semitone errors were separated from the group that committed more broad errors. More research is necessary to examine the causes for the differences in the underlying models” . This can be misunderstood. I suggest writing: “Using the perfect scoring approach, 1.1% of participants had all items correct. According to the IRT model, these participants would be expected to have greater skills in isolated pitch recognition tasks than participants with lower numbers of correct responses. In contrast, for the imperfect scoring approach, the LCA model assumes that 20.9% of participants have high skills in isolated pitch recognition tasks. Within this group further differentiation in skills cannot be made. The 4.9% who had all 10 responses correct using the imperfect scoring approach were just luckier than the remaining 16% in the high-skill group. More research is necessary to examine the causes for the differences in the underlying models.”

Reviewer #2: The authors have addressed all of my comments, and I would like to thank them for considering the suggestions of the reviewers.

I have no further amendments to request, except some minor editing changes as listed below:

p.8

R is now mentioned – does this need a reference or software version number?

Line 308 states:

‘Table 4 shows the items level fit.’

Suggest this is changed to:

‘Table 4 shows item-level fit’

Line 324 states: ‘This table provides the item level for each item...’

Suggest this is changed to:

‘This table provides the item-level fit values for each item...’

Lines 340-342 state:

‘Considering the perfect approach, the lowest BIC was in favor of an IRT with two parameters. However, for the imperfect approach, the lowest BIC was in favor of an IRT with three parameters.’

Suggest this is changed to:

Considering the perfect approach, the lowest BIC was in favor of an IRT model with two parameters. However, for the imperfect approach, the lowest BIC was in favor of an IRT model with three parameters.

Line 345 states:

‘Therefore, for both perfect and imperfect models, we concluded that the two-parameters models fit better than three-parameters models’

Suggest this is changed to:

Therefore, for both perfect and imperfect models, we concluded that the two-parameter model fits better than three-parameter model.

Line 404 states:

‘Notably, the red group did not achieve 1, indicating a 100% probability of answering correctly for a giving stimulus’

Suggest this is changed to:

‘Notably, even the red group did not achieve a value of 1 for any of the items, which would indicate a 100% probability of answering correctly for a given stimulus’

Line 409 states:

‘This indicates that the ability to recognize isolated pitches in different timbres and registers without reference is better modeled as a continuous ability when the perfect rating approach is considered in comparison with a categorical and hybrid model’

Suggest this is changed to:

‘This indicates that the ability to recognize isolated pitches in different timbres and registers without reference is better modeled as a continuous ability, rather than when the perfect rating approach is considered with either a categorical or a hybrid model’

Line 446 states:

‘Moreover, in the perfect approach for two parameters, all the items showed high values of discrimination.'

Suggest this is changed to:

‘Moreover, for the two-parameter model of the perfect approach, all the items showed high values of discrimination.'

Line 490 states:

‘Interestingly, we observed that even the group classified as showing a high probability of choosing the correct answer (less than 20% of the 783 participants) across all the stimuli did not display 100% probability of answering correctly.’

Suggest this is changed to:

‘Interestingly, we observed that none of the individual stimuli were answered correctly 100% of the time, even among the group classified as showing a high probability of choosing the correct answer (less than 20% of the 783 participants).’

7. PLOS authors have the option to publish the peer review history of their article (what does this mean?). If published, this will include your full peer review and any attached files.

Reviewer #1: **Yes: **Jakob Bue Bjørner

Reviewer #2: No

---

## [Editor Report · Acceptance letter]

12 Feb 2021

PONE-D-20-04410R2 

A new approach to measuring absolute pitch on a psychometric theory of Isolated Pitch Perception: Is it disentangling specific groups or capturing a continuous ability? 

Dear Dr. Germano:

I'm pleased to inform you that your manuscript has been deemed suitable for publication in PLOS ONE. Congratulations! Your manuscript is now with our production department. 

Kind regards, 

on behalf of

Dr. Karl Bang Christensen 

Academic Editor

PLOS ONE